# CaMKII controls neuromodulation via neuropeptide gene expression and axonal targeting of neuropeptide vesicles

**Alessandro Moro**[1,2], **Geeske M. van Woerden**[3], **Ruud F. Toonen**[2]*,
**Matthijs Verhage**[1,2]*

1 Department of Clinical Genetics, Center for Neurogenomics and Cognitive Research (CNCR), University Medical Center Amsterdam, Amsterdam, the Netherlands, 2 Department of Functional Genomics, Center for Neurogenomics and Cognitive Research (CNCR), Vrije Universiteit (VU) Amsterdam, Amsterdam, the Netherlands, 3 Department of Neuroscience, ENCORE Expertise Center for Neurodevelopmental Disorders, Erasmus MC, Rotterdam, the Netherlands

* matthijs@cncr.vu.nl (MV); ruud.toonen@cncr.vu.nl (RFT)

**Data Availability Statement:** All relevant data are within the paper and its Supporting Information files.

## Abstract

$Ca^{2+}$/calmodulin-dependent kinase II (CaMKII) regulates synaptic plasticity in multiple ways, supposedly including the secretion of neuromodulators like brain-derived neurotrophic factor (BDNF). Here, we show that neuromodulator secretion is indeed reduced in mouse α- and βCaMKII-deficient (αβCaMKII double-knockout [DKO]) hippocampal neurons. However, this was not due to reduced secretion efficiency or neuromodulator vesicle transport but to 40% reduced neuromodulator levels at synapses and 50% reduced delivery of new neuromodulator vesicles to axons. αβCaMKII depletion drastically reduced neuromodulator expression. Blocking BDNF secretion or BDNF scavenging in wild-type neurons produced a similar reduction. Reduced neuromodulator expression in αβCaMKII DKO neurons was restored by active βCaMKII but not inactive βCaMKII or αCaMKII, and by CaMKII downstream effectors that promote cAMP-response element binding protein (CREB) phosphorylation. These data indicate that CaMKII regulates neuromodulation in a feedback loop coupling neuromodulator secretion to βCaMKII- and CREB-dependent neuromodulator expression and axonal targeting, but CaMKIIs are dispensable for the secretion process itself.

## Introduction

Neuropeptides and neuromodulators are secreted via $Ca^{2+}$-dependent fusion of dense-core vesicles (DCVs) found in axons and dendrites of most neurons in the brain. Upon binding to their receptors, the secreted molecules modulate synaptic function [1, 2] and synaptic plasticity. One of the well-documented examples is how brain-derived neurotrophic factor (BDNF) facilitates long-term potentiation (LTP) and learning [3, 4]. However, the mechanism by which molecules like BDNF are secreted from DCVs is still a poorly defined element. DCV exocytosis is strictly $Ca^{2+}$-dependent and requires higher frequency stimulation than synaptic

**Funding:** This work is supported by an ERC Advanced Grant to MV. FP7-IDEAS-ERC ID:322966 https://cordis.europa.eu/project/rcn/107066/factsheet/en The funders had no role in study design, data collection and analysis, decision to publish, or preparation of the manuscript.

**Competing interests:** The authors have declared that no competing interests exist.

**Abbreviations:** AIS, axon initial segment; AMPA, alpha-Amino-3-hydroxy-5-methyl-4-isoxazolepropionic acid; BDNF, brain-derived neurotrophic factor; BDNF2, BDNF transcript II; BDNF Ab, BDNF antibody; BFP, blue fluorescence protein; CaMK, Ca2+/calmodulin-dependent kinase; CHGB, chromogranin B; CREB, cAMP-response element binding protein; CREB-Y134F, phosphorylation of CREB with Tyr-to-Phe substitution at position 134; CREB-S133A, CREB with Ser-to-Ala substitution at position 133; DCV, dense-core vesicle; DIV, day in vitro; DKO, double-knockout; EPSC, excitatory postsynaptic current; F-actin, filamentous actin; GFP, green fluorescence protein; KD, kinase-dead; KIF, kinesin; LA, long-active; LTD, long-term depression; LTP, long-term potentiation; MAPK, mitogen-activated protein kinase; mGFP, membrane-bound GFP; MUNC18, mammalian uncoordinated (UNC) 18; Nav$_{II-III}$, voltage-gated sodium channel intracellular domain; NLS-Nrgn, nuclear-localized neurogranin; NMDA, N-methyl-D-aspartate; NMJ, neuromuscular junction; NPY, neuropeptide Y; pCREB, CREB phosphorylation; PSD, postsynaptic density; SNAP25, synaptosomal-associated protein 25; SNARE, soluble N-ethylmaleimide-sensitive factor (NSF) attachment protein receptor; SV, synaptic vesicle; SypHy, synaptophysin-pHluorin; Syn1, synapsin 1; Syt1, synaptotagmin 1; TeNT, tetanus neurotoxin; TetON, tetracycling inducible expression system; TrfR, transferrin receptor; TrkB, tyrosine receptor B; VAMP, vesicle-associated membrane protein; VGLUT1, vesicular glutamate transporter 1; WT, wild type.

vesicle exocytosis [5, 6]. Several studies in invertebrates [7–9] and one in mouse cortical neurons [10] have proposed that Ca$^{2+}$/calmodulin-dependent protein kinase 2 (CaMKII) is involved in DCV exocytosis. This would be consistent with the fact that both secretion of neuromodulators and activation of the kinase [11, 12] requires high-frequency stimulation. Hence, CaMKII may, in concert with its firmly established role in LTP-induction, upstream of alpha-Amino-3-hydroxy-5-methyl-4-isoxazolepropionic acid (AMPA) receptor insertion and–phosphorylation [13–15], also be responsible for neuromodulator secretion from DCVs to contribute to plasticity phenomena like LTP.

Of the 4 genes that encode CaMKII proteins, α and βCaMKII are the most abundant isoforms expressed in neurons [16–18] and share more than 90% homology in their catalytic domain [17]. Generally, βCaMKII has a nonenzymatic function binding to filamentous actin (F-actin) and positioning the holoenzyme, together with αCaMKII, at the postsynaptic density (PSD) after Ca$^{2+}$ influx through N-methyl-D-aspartate (NMDA) receptors [19–21]. Once activated, αCaMKII initiates a phosphorylation cascade with targets from AMPA receptors [12, 22], to motor proteins [23] and cytoskeleton components [24]. In addition, γCaMKII shuttles CaM to the nucleus promoting CaMKIV activation and therefore indirectly regulates transcription via phosphorylation of cAMP-response element binding protein (CREB) [25–27], which promotes the transcription of BDNF [28–30]. At the presynaptic side, CaMKII regulates presynaptic plasticity [31] by limiting the ready releasable pool of synaptic vesicles (SVs) [32]. In *Drosophila* neuromuscular junction (NMJ), pharmacological inhibition of CaMKII reduces both the mobility of DCVs inside synaptic boutons [7, 33] and the synaptic capture of DCVs traveling through the axon [8, 34], causing failures in neuropeptide secretion after high-frequency stimulation [7]. In mice, similar pharmacological inhibition delays BDNF exocytosis in dendrites [10]. In *Caenorhabditis elegans*, loss of the CaMKII homolog UNC-43 leads to a drastic reduction of neuropeptides at synapses, but their accumulation in coelomocytes was unaltered [9], i.e., suggesting that secretion, per se, opposite to what was reported in *Drosophila*, was not affected. Hence, despite some contrasting findings, these studies suggest that CaMKII is involved in efficient positioning and possibly also the release of neuropeptides and neuromodulators from DCVs.

In this study, α- and βCaMKII double-knockout (DKO) hippocampal neurons were used to investigate the role of CaMKII in neuropeptide and neuromodulator trafficking and exocytosis. Using a combination of cellular and single-vesicle approaches, we confirmed that CaMKII indeed regulates the amount of secreted neuromodulators. However, we found that the efficiency of individual DCV fusion events was unaffected and that the fraction of fusing vesicles remained the same in the absence of α- and βCaMKII. Instead, we discovered that the reduced secretion of neuromodulators was entirely due to a decreased neuropeptide/neuromodulator content in synapses and a decreased number of DCVs entering the axons of DKO neurons. The decreased neuropeptide content was explained by a robust and selective reduction in the expression of DCV cargo in DKO neurons. Blocking BDNF secretion or antibody scavenging of secreted BDNF in wild-type (WT) neurons produced a similar reduction. Reduced neuromodulator expression in αβCaMKII DKO neurons was restored by active βCaMKII, but not inactive βCaMKII or αCaMKII, and by 3 CaMKII downstream effectors that promote CREB phosphorylation (pCREB). We conclude that in mammalian CNS neurons, CaMKII regulates neuromodulation as a crucial component of a feedback loop that up-regulates expression of DCV-resident neuromodulators dependent on their secretion. In addition, CaMKII regulates axonal targeting of neuromodulators but not DCV transport or fusion.

## Results

### CaMKII facilitates neuropeptide secretion

To understand the role of CaMKII in neuropeptide secretion, we used αCaMKII and βCaMKII conditional DKO mice [35]. Hippocampal neuron cultures were infected with Cre-recombinase (DKO) or an inactive form of Cre [36] (WT) on the day of plating (day in vitro, DIV0) (Fig 1A), producing complete knockout of both α and βCaMKII at DIV17 (Fig 1B). To investigate DCV fusion, we performed live-cell imaging of the DCV cargo BDNF-pHluorin [37] (Fig 1C), which allowed us to probe neuropeptide exocytosis at single-vesicle resolution (Fig 1D and S1 Movie). Neurons were stimulated twice with 8 bursts of 1 second at 50 Hz to trigger DCV fusion [6], and individual BDNF-pHluorin fusion events (Fig 1E) were counted. The number of BDNF-pHluorin fusion events was lower in DKO neurons compared with WT (WT = 515 ± 281.11; DKO = 282 ± 153.15 $p$ = 0.0575; Fig 1G). This reduction indicates that neuromodulators, such as BDNF, require CaMKII for efficient secretion, as suggested in *Drosophila* NMJ [7] and by pharmacological inhibition in mouse [10].

The majority of DCVs fuse at synapses [38, 39]. In *Drosophila*, CaMKII inhibition reduces the amount of DCV fusion at the NMJ synapse [7]. We investigated the localization of BDNF-pHluorin fusion events (Fig 1H) and observed no differences in the percentage of BDNF-pHluorin fusion events that occurred at Synapsin-mCherry marked synapses (WT = 68.18% ± 4.42; DKO = 71.81% ± 3.54 $p$ = 0.106; Fig 1I). Thus, CaMKII facilitates the secretion of neuromodulators both at synaptic as well as at extrasynaptic locations.

To assess the specificity for neuropeptide transmission, we used Synaptophysin-pHluorin (SypHy) [40] to analyze SV transmission (S1 Fig). SypHy fluorescence was quantified during electrical field stimulation of 5 s at 40 Hz, which activated the majority of synapses (WT = 95.05 ± 4.60; DKO = 93.04 ± 3.56 $p$ = 0.4431; S1 Fig), containing equal levels of SypHy-positive vesicles (S1 Fig). As suggested in previous studies in αCaMKII KO [32] as well as α- and βCaMKII DKO [22, 35], increase in SypHy signal during stimulation (S1 Fig), as well as its decay during the recovery period (S1 Fig), was comparable between DKO and WT neurons. We, therefore, concluded that CaMKII contributes to neuropeptide exocytosis specifically over synaptic transmission.

### CaMKII affects the number of DCVs, not their fusion

To investigate the mechanism of reduced BDNF secretion, we measured the fraction of DCVs that fused upon stimulation [6]. Strikingly, the total number of pHluorin-labeled DCVs, quantified upon brief $NH_4Cl$ superfusion (Fig 2A), was reduced by 43% in DKO neurons compared with WT (WT = $7.72 \times 10^3$ ± $1.19 \times 10^3$; DKO = $4.44 \times 10^3$ ± 386; $p$ = 0.0124; Fig 2B). The same effect is also observed as a leftward shift in the histogram of total DCVs/per cell (S1 Fig). The reduction in pHluorin-labeled DCVs was not due to a defect in acidification of the vesicles, because the baseline fluorescence was comparable in the 2 genotypes (WT = 1 ± 0.017; DKO = 0.961 ± 0.019; $p$ = 0.1437 S2 Fig). Taking into account the lower number of pHluorin-labeled DCVs, the fraction of DCVs fusing in DKO neurons that was similar to WT (WT = 6.83% ± 2.4; DKO = 5.6% ± 2.2; $p$ = 0.3575; Fig 2C and 2D), indicating that CaMKII does not control the DCV fusion process itself but regulates the number of available DCVs.

The reduced signal upon $NH_4Cl$ superfusion could be due to reduced loading of BDNF-pHluorin into vesicles. The increase in signal intensity during fusion was calculated to estimate the loading of BDNF-pHluorin into vesicles (S1 Fig). In the 2 conditions, the fusion intensity was similar (WT = 1.494 ± 0.101; DKO = 1.519 ± 0.033; $p$ = 0.748; S2 Fig). Hence, CaMKII is not involved in the loading of neuropeptides into DCVs.

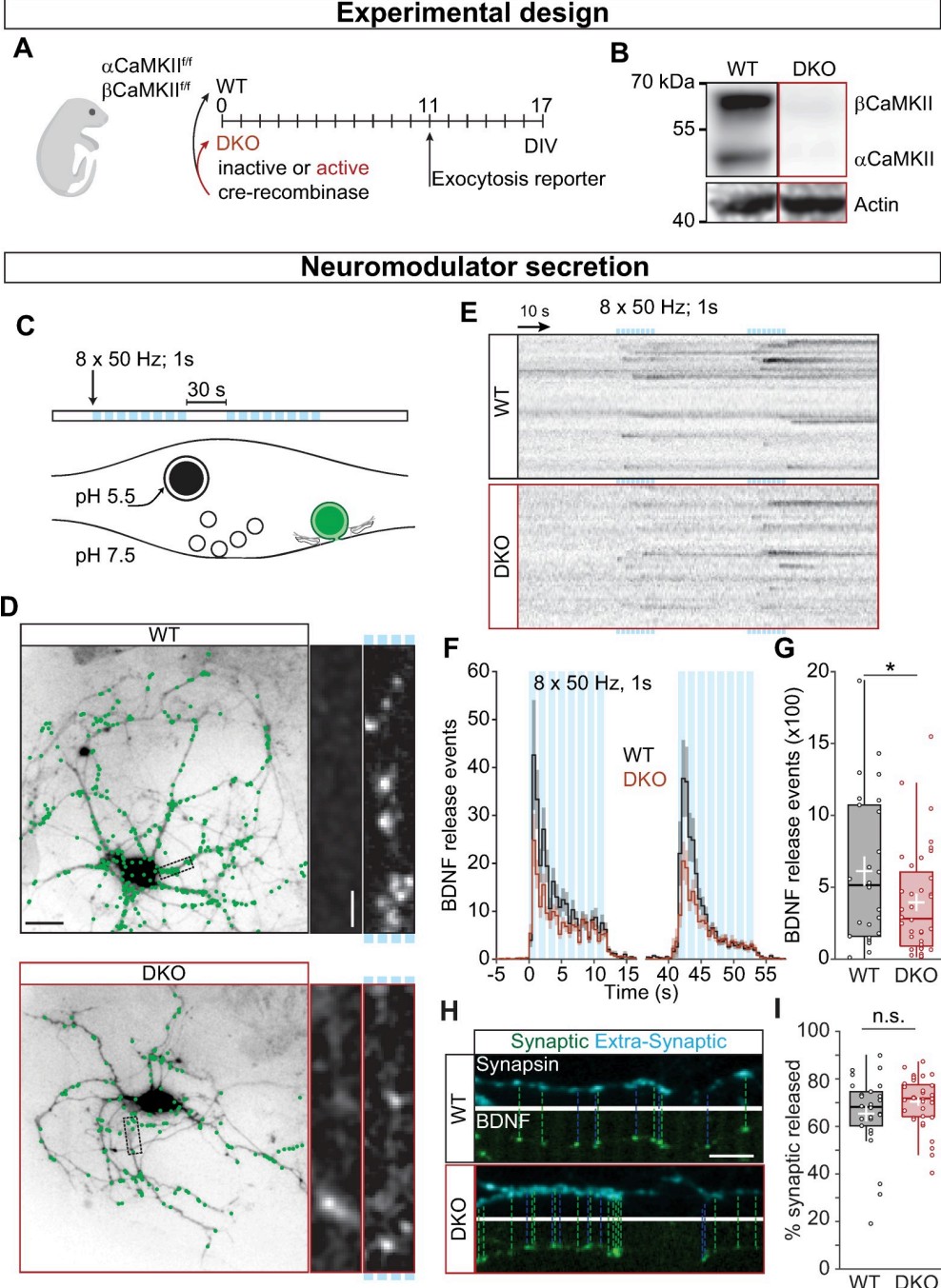

**Fig 1. CaMKII regulates neuromodulation.** (A) Schematic representation of the experimental design. Neuronal cultures from floxed αβCaMKII P1 pups were infected at DIV0 with Cre-recombinase to generate DKO neurons and with inactive Cre as control. Cultures were infected at DIV11 with BDNF-pHluorin as exocytosis reporter, and live-cell imaging was performed at DIV17. (B) Typical Western blot showing the complete KO for both α and βCaMKII. (C) Schematic representation of DCV fusion assay. DCVs are labeled with BDNF-pHluorin, and neurons are stimulated with 2 trains of 8 bursts of 50 APs at 50 Hz (light blue bars). (D) Representative neurons expressing CFP driven by CaMKII promoter superimposed with BDNF-pHluorin events (green dots). Insert shows a neurite at resting state (left) and upon stimulation (marked by light blue bars on the right). (E) Kymographs of BDNF-pHluorin fusion events. Individual DCV fusion events, black dots in the images, were quantified semi-automatically. (F) Histogram of the average number of DCV fusion events per cell during stimulation (light blue bars). (G) Total number of BDNF-pHluorin release events per cell. (H) Typical example of BDNF-pHluorin fusion events at synapses visualized by Synapsin-mCherry (dashed lines) and at extrasynaptic sites. (I) Percentage of BDNF-pHluorin fusion events at

synapses. Traces show mean ± SEM (shaded area); boxplots with 95% CI whiskers, the white cross represents mean ± SEM, and the central bar is median. Columns and dots represent individual litters and neurons, respectively. The presented data can be found in S1 Data; original Western blot can be found in S1 Raw images. *$p$ = 0.057. Scale bar = 25 μm (D); 5 μm (D insert); 10 μm (I). AP, action potential; BDNF, brain-derived neurotrophic factor; CaMKII, Ca2+/calmodulin-dependent kinase II; CFP, cyan fluorescence protein; CI, confidence interval; DCV, dense-core vesicle; DIV, day in vitro; DKO, double-knockout; WT, wild type.

Taken together, our data demonstrate that CaMKII is a positive regulator of neuromodulation by promoting the number of available DCVs. Moreover, CaMKII is not involved in the fusion of SVs.

## CaMKII DKO neurons have reduced number of DCVs per synapse

Because the total number of BDNF-pHluorin-labeled DCVs was significantly reduced (Fig 2B), we tested whether CaMKII regulates the expression of the reporter by regulating the promoter activity of the synapsin promoter in our reporter construct. Like many other studies, we

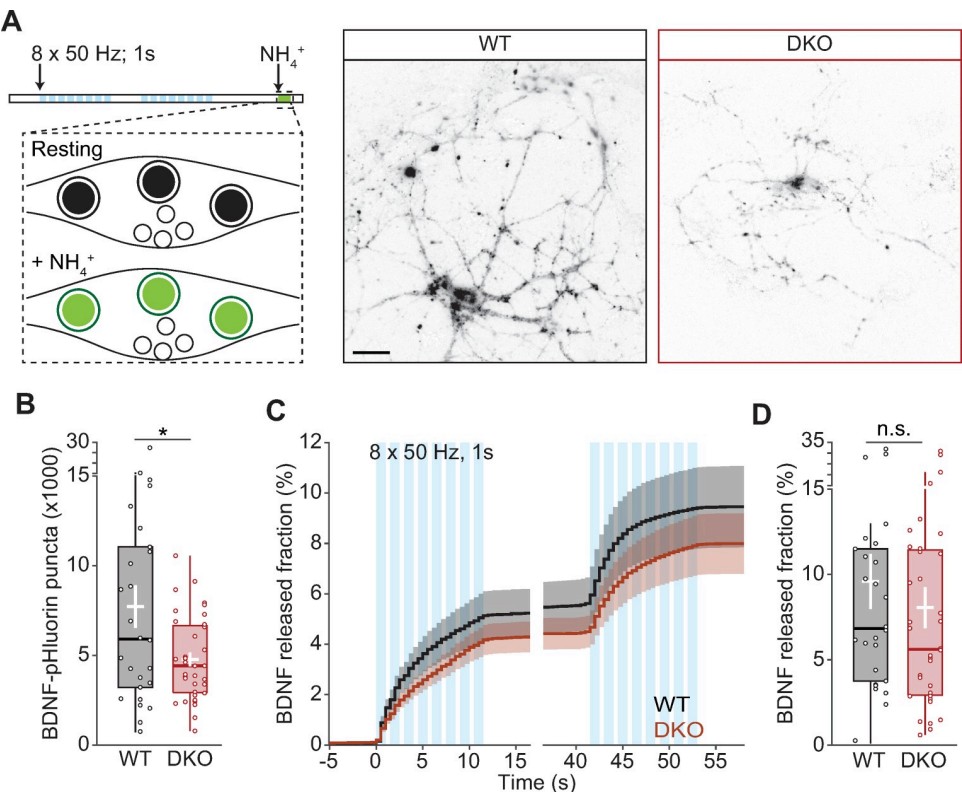

**Fig 2. CaMKII regulates DCV content but not their fusion efficiency.** (A) Schematic representation of DCV fusion assay. DCVs are labeled with BDNF-pHluorin, and neurons are stimulated with 2 trains of 8 bursts of 50 APs at 50 Hz (light blue bars). At the end of the stimulation, the bath application of $NH_4Cl$ dequenches all BDNF-pHluorin-containing vesicles. Right, representative neurons with BDNF-pHluorin-labeled vesicles dequenched. (B) Total number of DCVs per cell quantified upon $NH_4^+$ superfusion. (C) Cumulative fraction of DCV fusion events during stimulation. The fraction is calculated as the number of DCVs fusing per frame divided by the total number of BDNF-pHluorin puncta per neuron. (D) Fraction of BDNF-pHluorin-labeled DCV fusing during stimulation. Traces show mean ± SEM (shaded area); boxplots with 95% CI whiskers, the white cross represents mean ± SEM, and the central bar is median. Columns and dots represent individual litters and neurons, respectively. The presented data can be found in S1 Data. *$p$ < 0.05. Scale bar = 25 μm (A). AP, action potential; BDNF, brain-derived neurotrophic factor; CaMKII, Ca2+/calmodulin-dependent kinase II; CI, confidence interval; DCV, dense-core vesicle; DIV, day in vitro; DKO, double-knockout; WT, wild type.

designed these reporters to be specific for neurons and do not express in the supporting glia in the cultures, using the widely used (human) synapsin promoter. However, the expression levels of all constructs driven by this synapsin promoter were similar (S2 Fig).

Second, to better characterize the effects of CaMKII on the number of DCVs, neurons were immunostained for a series of endogenous and exogenous DCV markers (Fig 3A–3D). Both endogenous BDNF (WT = 0.920 ± 0.144; DKO = 0.535 ± 0.107; $p = 5.54 \times 10^{-4}$; Fig 3B) and Chromogranin B (CHGB) (WT = 0.895 ± 0.188; DKO = 0.421 ± 0.077; $p = 2.53 \times 10^{-7}$; Fig 3C), and exogenous neuropeptide Y (NPY) pHluorin (WT = 1.023 ± 0.130; DKO = 0.446 ± 0.094; $p = 8.41 \times 10^{-7}$; Fig 3D) levels were severely reduced at synapses. Because the loading of neuropeptides into vesicles was unchanged in DKO neuron (S1 Fig), these data indicate that the average number of DCVs at synapses was reduced in the absence of CaMKII.

To further investigate the lower neuromodulator levels in CaMKII DKO neurons, the number of DCVs was assessed along the neuritic arborization and stainings for endogenous neuromodulators were quantified as a function of the distance from the soma. The number of DCVs along the neuritic arborization was lower in CaMKII-deficient neurites (S2 Fig), and the staining intensity was reduced in CaMKII DKO neurons at every point in the dendritic arborization (S3 Fig). These data support our previous conclusion that CaMKII-deficient neurons contain fewer DCVs and, consequently, less neuromodulators in their neuritis. This suggests that the reduction in neuromodulator in CaMKII DKO neurons is explained by defects upstream of neurite targeting.

To evaluate whether this reduction is specific for DCVs, we analyzed the signal intensity of the SV markers vesicular glutamate transporter 1 (VGLUT1) (Fig 3F), Synaptobrevin 2 (vesicle-associated membrane protein 2 [VAMP2]) (Fig 3G) and Synapsin 1 (Syn1) (Fig 3H). The levels of endogenous VGLUT1 and Syn1 were unaltered in the 2 conditions, in line with the SypHy (S1 Fig) and Synapsin-mCherry data (S2 Fig). We observed a minor reduction in the levels of VAMP2 (WT = 0.985 ± 0.087; DKO = 0.768 ± 0.129; $p = 0.009$). Moreover, in CaMKII-deficient synapses, the levels of mammalian uncoordinated (UNC) 18 (MUNC18-1) (WT = 0.941 ± 0.098; DKO = 0.656 ± 0.104; $p = 7.65 \times 10^{-5}$; Fig 3J) and synaptosomal-associated protein 25 (SNAP25) (WT = 0.929 ± 0.114; DKO = 0.712 ± 0.088; $p = 0.0017$; Fig 3K) were also decreased, whereas Synaptotagmin1 (Syt1) (Fig 3L) was unaltered. These changes in VAMP2, MUNC18, and SNAP25 levels do not result in reduced SV fusion (S2 Fig) nor to reduced synaptic transmission in brain slices of CaMKII DKO mice [35]. These findings suggest that CaMKII has a minor role in regulating the accumulation of SV-associated and exocytic proteins.

We then reasoned that the combined effect of CaMKII loss and neurotrophin reduction could lead to morphological defects. It is known that CaMKII limits dendritic arborization by regulating Rem2-GTPase activity and by regulating ubiquitin signaling to the centrosome [41, 42]. Indeed, dendrite length was increased in CaMKII DKO neurons (WT = 4.83 ± 0.7 mm; DKO = 7.036 ± 0.958 mm; $p = 0.00115$; S4 Fig). Surprisingly, the axonal length of DKO neurons was also increased (WT = 13.46 ± 3.02 mm; DKO = 23.52 ± 3.37 mm; $p = 0.00248$; S4 Fig), contrary to what was previously reported in *Drosophila* larvae [43]. The increase in neurite length was accompanied by an increased synapse density (WT = 25.5 ± 1.6 synapses per mm; DKO = 27.036 ± 0.958; $p = 0.05$; S4 Fig). This indicates that CaMKII limits neurite growth and synaptogenesis in autaptic cultures.

These results indicate that CaMKII is required to maintain normal neuropeptide levels in synapses, as well as some members of the regulated secretory machinery, hence indirectly affecting neuropeptide secretion.

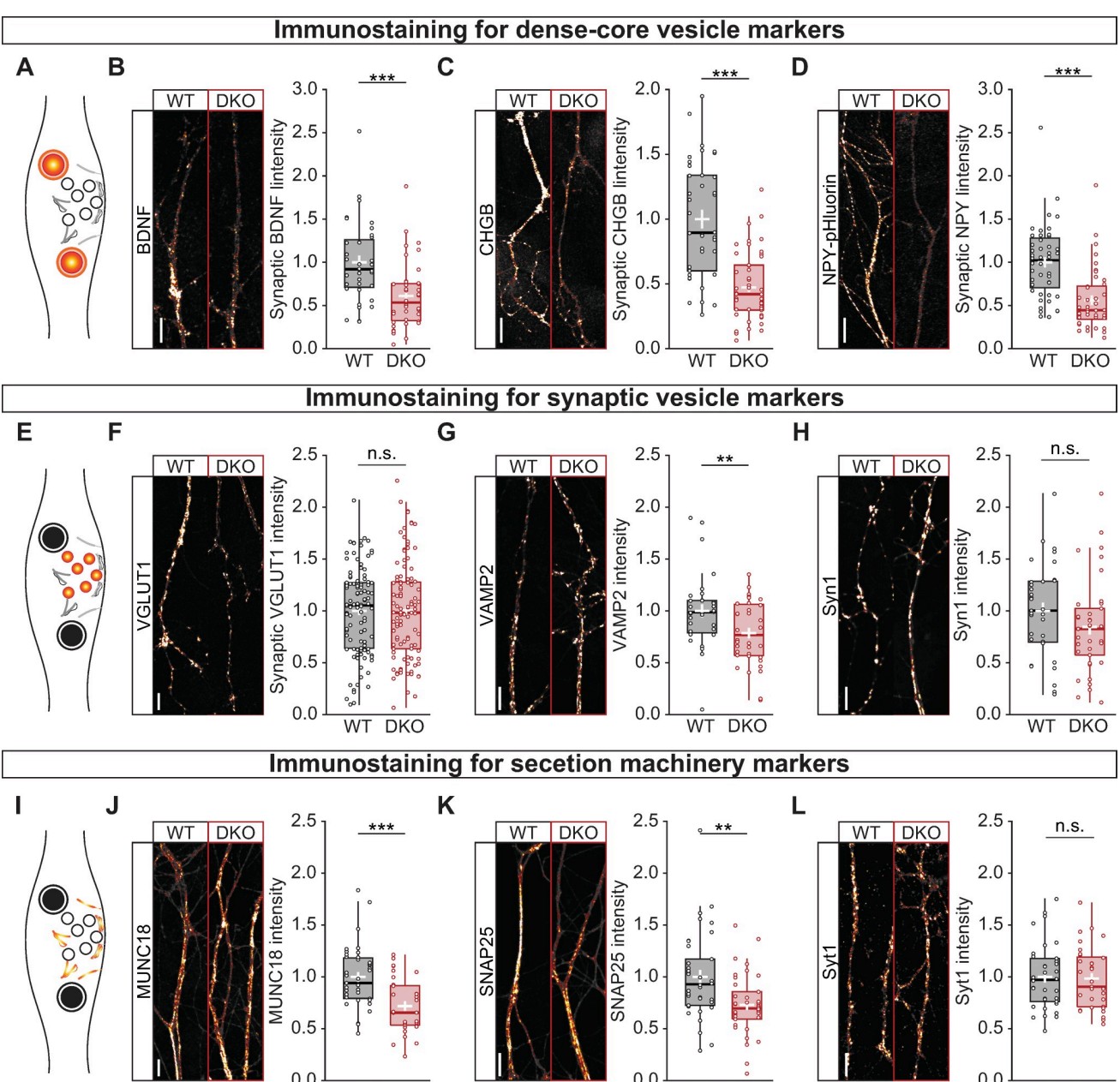

**Fig 3. CaMKII controls the abundance of DCVs, not SVs or secretory machinery.** (A) Schematic representation of the immunostaining of DCV markers, in color. (B) Left, typical images of neurites of WT (left) and DKO (right) immunostained for BDNF. Right, quantification of BDNF intensity at VGLUT1 labeled synapses in WT and DKO neurons. (C) Left, typical images of neurites of WT (left) and DKO (right) immunostained for CHGB. Right, quantification of CHGB intensity at VGLUT1 labeled synapses in WT and DKO neurons. (D) Left, typical images of neurites of WT (left) and DKO (right) immunostained for the exogenous DCV marker NPY-pHluorin. Right, quantification of NPY-pHluorin intensity at VGLUT1 labeled synapses in WT and DKO neurons. (E) Schematic representation of the immunostaining of SV markers, in color. (F) Left, typical images of neurites of WT (left) and DKO (right) immunostained for VGLUT1. Right, quantification of VGLUT1 intensity in WT and DKO neurons. (G) Left, typical images of neurites of WT (left) and DKO (right) immunostained for VAMP2. Right, quantification of VAMP2 intensity at VGLUT1-labeled synapses in WT and DKO neurons. (H) Left, typical images of neurites of WT (left) and DKO (right) immunostained for Syn1. Right, quantification of Syn1 intensity at VGLUT1 labeled synapses in WT and DKO neurons. (I) Schematic representation of the immunostaining of exocytic proteins, in color. (J) Left, typical images of neurites of WT (left) and DKO (right) immunostained MUNC18. Right, quantification of MUNC18 intensity at VGLUT1-labeled synapses in WT and DKO neurons. (K) Left, typical images of neurites of WT (left) and DKO (right) immunostained for SNAP25. Right, quantification of SNAP25 intensity at VGLUT1 labeled synapses in WT and DKO neurons. (L) Left, typical images of neurites of WT (left) and DKO (right) immunostained for Syt1. Right, quantification of Syt1 intensity at VGLUT1-labeled synapses in WT and DKO neurons. Boxplots with 95% CI whiskers, white cross shows mean ± SEM, central bar is median. Columns and dots represent individual litters and neurons, respectively. The presented data can be found in S1 Data. $*p < 0.05$, $**p < 0.01$, $***p < 0.001$. Scale bar = 5 μm. CaMKII, Ca2+/calmodulin-dependent kinase II; CI, confidence interval; DCV, dense-core vesicle; DKO, double-

knockout; MUNC18, mammalian uncoordinated 18; NPY, neuropeptide Y; n. s., not significant; SNAP25, synaptosomal-associated protein 25; SV, synaptic vesicle; Syn1, synapsin 1; VAMP2, vesicle-associated membrane protein 2; VGLUT1, vesicular glutamate transporter 1; WT.

## Fewer DCVs enter the axon of CaMKII DKO neurons, whereas DCV trafficking is unaffected

To evaluate whether the reduced neuropeptide levels at synapses is a consequence of a decreased loading of neuropeptides into DCVs or a decreased number of DCVs entering the axon, hippocampal neurons expressing BDNF-mCherry and the axon initial segment (AIS) marker voltage-gated sodium channel intracellular domain fused to blue fluorescent protein (BFP-NaV$_{II-III}$) [44] (Fig 4A) were photobleached at the location of the AIS to visualize DCV trafficking into the axon (Fig 4B and S2 Movie). The number of BDNF-labeled vesicles trafficking into the AIS in CaMKII DKO neurons was reduced by 42% (WT = 4.115 ± 0.257 DCV per minute; DKO 2.395 ± 0.171; $p = 1.52 \times 10^{-7}$; Fig 4C), in line with the reduction in intensity levels of endogenous BDNF and CHGB at synapses. To test whether the decreased number of trafficking DCVs was specific for the axon, 2 dendritic regions proximal to the soma were photobleached (Fig 4E). The targeting of DCVs into dendrites was similar in WT and DKO neurons (Fig 4F). The reduction in axonal targeting but normal dendritic targeting together leads to decreased axonal preference of DCV targeting (WT = 68.9 ± 0.05%; DKO = 60.6 ± 0.06; $p = 0.0033$; Fig 4G). In addition, the number of BDNF-labeled DCVs traveling in a retrograde direction was increased by 35% (WT = 1.5 ± 0.29 DCV per minute; DKO = 2.0 ± 0.43; $p = 0.0155$; S5 Fig) in DKO neurons. Further analysis on the run length and pausing time of DCVs at the AIS did not show any difference between WT and CaMKII DKO neuron (S5 Fig), suggesting that the once the DCVs are loaded to the microtubles, their trafficking is CaMKII independent.

Extracellular, released BDNF is known to be transported to the soma when associated with tyrosine receptor B (TrkB) and subsequently internalized [45]. We observed that CaMKII DKO neurons have increased TrkB internalization (WT = 77.0 ± 5.3 puncta per mm; DKO = 84.5 ± 3.8; $p = 0.037$; S5 Fig); however, extracellular TrkB intensity, as well as extracellular transferrin receptor (TrfR), was unaltered in CaMKII DKO neurons compared with WT neurons (S5 Fig) indicating that, at steady state, there is no defect in trafficking and secretion of constitutive secreted vesicles. DCV trafficking speed in both anterograde (Fig 4D) and retrograde (S5 Fig) direction was on average 1 μm/s and similar between DKO and WT neurons. Hence, CaMKII regulates neuropeptide content in synapses by facilitating the transport from the soma into the axon of hippocampal neurons.

In *Drosophila* NMJ, CaMKII regulates DCV mobilization and capture at synapses [7, 8]. Mammalian DCVs stall during high-frequency stimulation [46], a process that might involve a CaMKII-dependent modification of the cargo-kinesins (KIFs) complex, like for the interaction of Mint1 and KIF17 [23]. To evaluate this, we used high-density hippocampal cultures with 5% of neurons expressing tetracycling inducible (TetON) membrane-bound green fluorescence protein (mGFP) and NPY-mCherry. This allowed for discrimination between axon and dendrites to study DCV trafficking in both compartments (Fig 4H and S3 Movie). DCV speed was measured during high-frequency stimulation of 16 repetitions of 1-second field stimulation at 50 Hz and compared with speed during baseline (30 seconds before stimulation). The speed of moving DCVs was reduced during stimulation (Fig 4I) for approximately 40% of vesicles in both anterograde and retrograde direction in both genotypes (S5 Fig). The fraction of vesicles that stalled (speed reduced to zero) was 16%–18%, whereas 24%–29% increased their speed during stimulation, similar in both genotypes (Fig 4J). Therefore, CaMKII promotes the

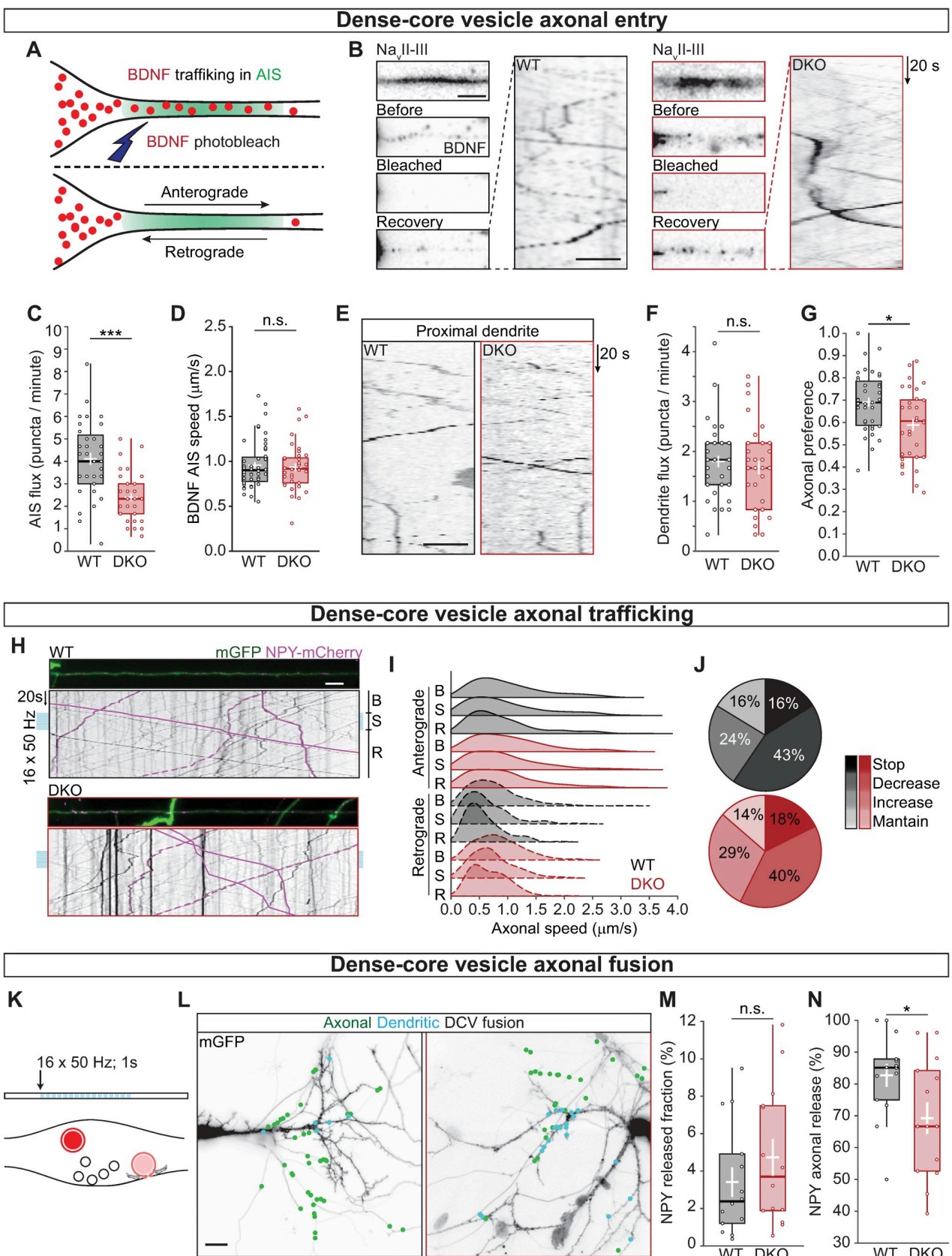

**Fig 4. CaMKII regulates the number of DCVs entering the axon but not their trafficking properties.** (A) Schematic representation of AIS targeting; Na$_v$II-III BFP was used to visualize the region of the AIS. BDNF-mCherry vesicles were bleached at the AIS to allow quantification of new vesicles entering the axon. (B) Left, typical example of AIS (top) with BDNF-mCherry-labeled DCVs before bleaching, after bleaching, and at the end of the recovery (top to bottom). Right, kymograph showing DCVs in the AIS after bleaching. (C) Quantification of DCV flux at the AIS calculated as the number of BDNF-mCherry positive puncta that enter the Na$_v$II-III BFP area per minute in anterograde (from the soma to the axon) direction. (D) Quantification of BDNF-mCherry speed at the AIS in the anterograde direction. (E) Kymograph showing DCVs in proximal dendrites after bleaching. (F) Quantification of DCV flux in the proximal dendrites in the anterograde direction. (G) Axonal preference of DCV targeting, calculated as BDNF-mCherry-positive puncta entering in the Na$_v$II-III BFP area in anterograde direction divided by the total number of BDNF-mCherry positive puncta exiting the soma. (H) Typical examples of axons labeled with mGFP showing NPY-mCherry labeled DCVs; kymographs represent axonal trafficking of NPY-mCherry in anterograde (solid lines) and retrograde (dashed lines) direction. Neurons were stimulated with 16 trains of 50 APs at 50 Hz (light blue bars). DCV speed was calculated before stimulation (B), during stimulation (S), and during recovery (R). (I) Ridgelines representing the density function of NPY-mCherry puncta speed along the axon, showing the characteristic differences in speed for anterograde and retrograde (filled and dashed density respectively) and the lower speed during stimulation. (J) Pie charts for WT (black) and DKO neurons (red) with the fraction of vesicles that stopped during stimulation (Stop), decreased their speed (Decrease), increased their speed (Increase), and maintained the same average speed (Maintain). (K) Schematic representation of DCV fusion assay. DCVs are labeled with NPY-mCherry, neurons are stimulated with 16 trains of 50 APs at 50 Hz (light blue bars). Individual DCV fusion events were quantified manually as puncta that suddenly lost their fluorescence signal. (L) Typical neurons labeled with mGFP overlaid with NPY-mCherry-labeled DCV fusion events (green dots). Axons were identified by the lack of spines. (M) Quantification of NPY-mCherry release fraction upon stimulation. (N) Quantification of the percentage of NPY-mCherry fusion events at axons. Boxplots with 95% CI whiskers, white cross shows mean ± SEM, and central bar is median. Columns and dots represent individual litters and neurons, respectively. The presented data can be found in S1 Data. $^*p < 0.05$, $^{**}p < 0.01$, $^{***}p < 0.001$. Scale bar = 5 μm (B for still frames), 10 μm (B kymograph and E), 25 μm (H), 25 μm (L). AIS, axon initial segment; AP, action potential; BDNF, brain-derived neurotrophic factor; BFP, blue fluorescence protein; CaMKII, Ca2+/calmodulin-dependent kinase II; CI, confidence interval; DCV, dense-core vesicle; DKO, double-knockout; mGFP, membrane-bound green fluorescence protein; Na$_{v\ II-III}$,voltage-gated sodium channel intracellular domain; NPY, neuropeptide Y; n. s., not significant; WT, wild type.

axonal targeting of DCVs. However, CaMKII is dispensable for a normal trafficking and stimulation-dependent arrest of trafficking vesicles in the axon.

In cultured hippocampal neurons, DCV fusion occurs predominantly in the axon [6]. To test whether this preference is altered in CaMKII DKO neurons, the location of DCV fusion (Fig 4K) was quantified in the same sparse infected neurons (Fig 4L). The total release fraction of NPY-mCherry-labeled DCVs was the same between the 2 groups (Fig 4M). However, the percentage of fusion that occurred in axons of αβCaMKII-deficient neurons was reduced by 22% (WT = 85.09 ± 5.39%; DKO = 66.67 ± 13.25%; $p$ = 0.0239; Fig 4N).

Together, these results show that CaMKII regulates the number of DCVs entering the axon and axonal preference of DCV fusion but not DCV trafficking properties.

## CaMKII regulates neuromodulator expression

The reduction of axonal preference for DCVs in CaMKII DKO neurons might be due to miss-targeting into dendrites or a general reduction in neuropeptide expression, thus reducing DCV number. Of the 4 BDNF transcripts [30, 47], transcript II and IV are regulated by low neuronal activity in a synergetic activity between CaMKII and PKC, via pCREB [28]. CREB binding sites are also present in the promoters of CHGB [48] and secretogranin2 (SCG2) [49]. Therefore, we reasoned that in our culture system CaMKII regulates transcription of neuro-peptide mRNAs (Fig 5A and 5B).

The mRNA levels of CHGB, SCG2, and BDNF transcript II (BDNF2) were significantly lower compared with WT (CHGB = 58%, SCG2 = 61%, BDNF2 = 50%; Fig 5B). BDNF transcript IV, on the other hand, was not significantly reduced (BDNF4 = 70%) yet was lower than WT levels. Expression of all transcripts was rescued by simultaneously overexpressing α and βCaMKII. Hence, in cultured hippocampal neurons, CaMKII regulates the transcription levels of DCVs cargos.

To evaluate that reduced transcription leads to the decreased axonal preference by reduced protein levels, we quantified total cellular protein levels by Western blot (Fig 5C) of the DCV markers CHGB and SCG2. The expression of both proteins was decreased in DKO neurons compared with WT neurons; CHGB levels were reduced to 60% of WT (Fig 5D) and SCG2 to

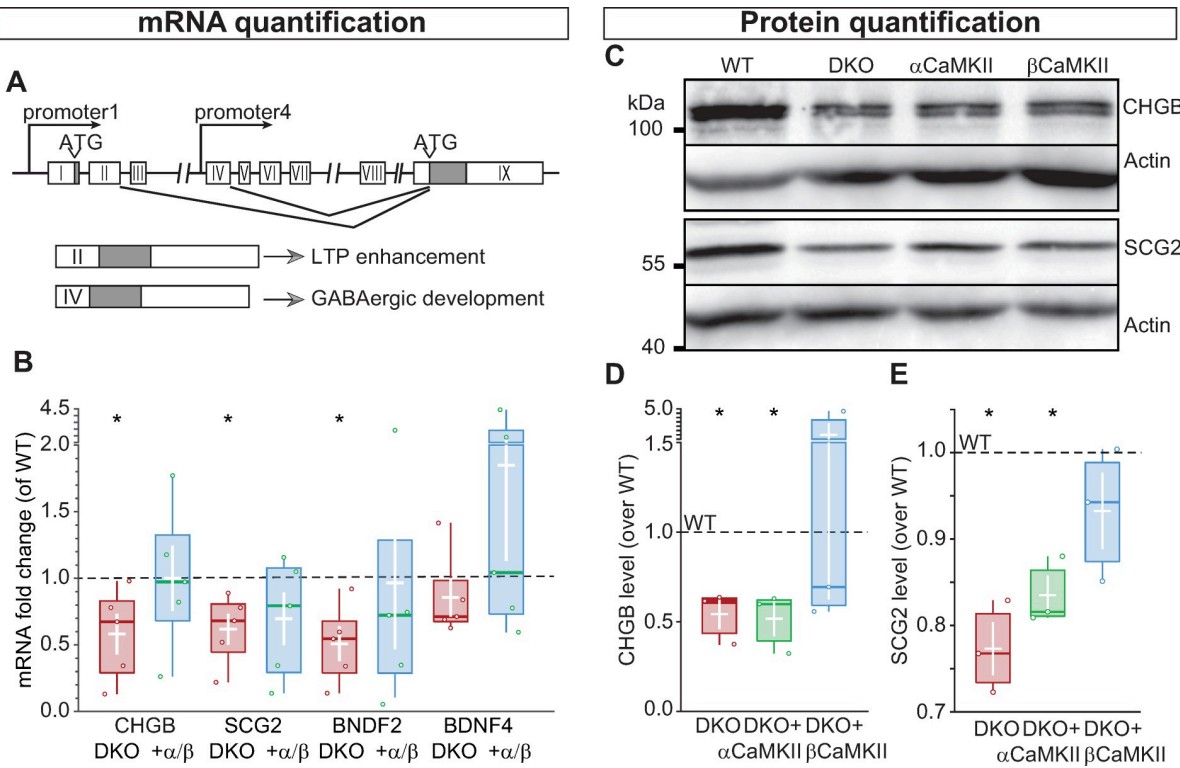

**Fig 5. CaMKII regulates neuropeptide expression.** (A) Representation of BDNF genomic locus, exons are shown as white boxes, the gray box represents the conserved exon shared between the different transcripts. (B) Quantification of mRNA fold change, expressed as $\Delta C_t$ values normalized to WT, showing the reduction of mRNA levels for CHGB, SCG2, and BDNF transcript 2 in DKO neurons compared with DKO neurons rescued with α and βCaMKII. BDNF transcript 4 is not affected by the lack of CaMKII. (C) Western blot showing the DCV cargo proteins CHGB (top) and SCG2 (bottom) in WT, DKO, and DKO-expressing αCaMKII or βCaMKII neurons. Actin was used as loading control. (D) Quantification of CHGB levels normalized to WT. (E) Quantification of SCG2 levels normalized to WT. Boxplots with 95% CI whiskers, white cross shows mean ± SEM, and central bar is the median. Columns and dots represent individual litters and neurons, respectively. The presented data can be found in S1 Data; original Western blot can be found in S1 Raw images. ATG, transcription starting codon; BDNF, brain-derived neurotrophic factor; CaMKII, Ca2+/calmodulin-dependent kinase II; CHGB, chromogranin B; CI, confidence interval; DCV, dense-core vesicle; DKO, double-knockout; LTP, long-term potentiation; SCG2, secretogranin2; WT, wild type.

76% (Fig 5E). CaMKII holoenzymes are multimers of α and β subunits and the action of the 2 subunits is interdependent, e.g., βCaMKII KO neurons have substantial impairments in αCaM-KII signaling, resembling αCaMKII KO [21]. Therefore, we overexpressed the individual subunits in the DKO background to test if either α- or βCaMKII has a dominant function in regulating neuropeptide levels in hippocampal neurons. αCaMKII did not restore neuropeptide levels, whereas βCaMKII restored expression levels to 70% and 95%, respectively, for CHGB and SCG2. Thus, βCaMKII regulates the transcription of neuromodulators, and the absence of the β gene is responsible for the reduction in the amount of neuromodulators produced.

## Secreted BDNF and CaMKII are required for BDNF-induced neuromodulator expression

BDNF is known to enhance transcription of neuropeptides and neuromodulators [50–52], including the precursor of BDNF itself [53–55]. In addition, extracellular, secreted BDNF is known to activate CaMKII via its receptor TrkB [56]. We therefore hypothesized that BDNF-dependent activation of CaMKII is required for a normal neuromodulator expression. To test this, BDNF was scavenged from the culture medium by adding 0.2 μg/ml of BDNF antibody (BDNF Ab) [57] at DIV 15 and DIV 16 and analyzed BDNF-induced BDNF expression. As a

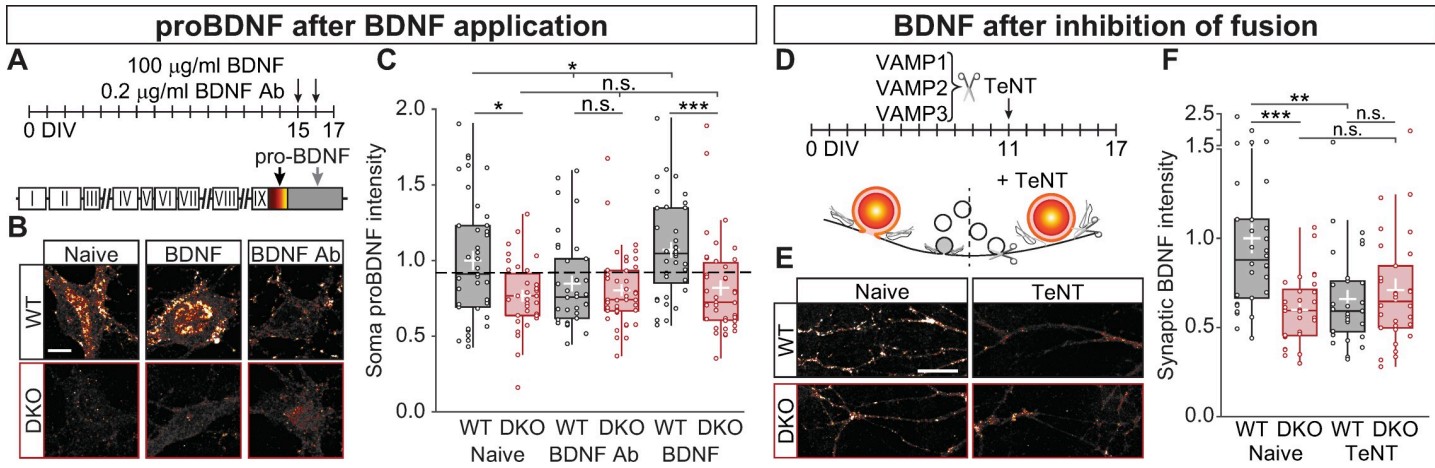

**Fig 6. Secreted BDNF is sufficient to increase BDNF levels in the presence of CaMKII.** (A) Schematic representation of the experimental design, 0.2 μg/ml of BDNF Ab or 100 ng/ml of recombinant BDNF were added every 24 hours for 2 days before fixing the neurons and immunostained for proBDNF. (B) Typical examples of proBDNF in the soma from untreated neurons (Naive), treated with BDNF Ab or BDNF. (C) Quantification of somatic proBDNF intensity normalized to WT (dash line). (D) Schematic representation of the experimental design, lentiviral particles containing TeNT, which is responsible to cleaved VAMP1-3 (hence, inhibiting SV and DCV fusion), were added 6 DIVs before fixing the neurons and immunostained for BDNF. (E) Typical examples of BDNF in synapses from untreated (Naive) cells or treated with TeNT. (F) Quantification of synaptic BDNF intensity normalized to WT (dash line). Boxplots with 95% CI whiskers, white cross shows mean ± SEM. Columns and dots represents individual litters and neurons, respectively. The presented data can be found in S1 Data. *$p < 0.05$, **$p < 0.01$, ***$p < 0.001$. Scale bar = 20 μm (B-E). BDNF, brain-derived neurotrophic factor; BDNF-Ab, BDNF antibody; CaMKII, Ca2+/calmodulin-dependent kinase II; CI, confidence interval; DCV, dense-core vesicle; DIV, day in vitro; DKO, double-knockout; SV, synaptic vesicle; TeNT, tetanus neurotoxin; VAMP, vesicle-associated membrane protein; WT, wild type.

control, 100 ng/ml of recombinant BDNF [58] was added at the same DIVs (Fig 6A). To avoid cross-contamination of the antibody and the recombinant BDNF, proBDNF was used for the quantification of CaMKII-dependent BDNF expression (Fig 6B). As expected, somatic proBDNF levels were reduced in CaMKII DKO neurons compared with WT neurons (WT = $1 ± 0.064$; DKO = $0.768 ± 0.039$; $p = 9.35 × 10^{-3}$; Fig 6C), in line with the reduced BDNF levels in synapses of DKO neurons (Fig 3B). When BDNF was scavenged, the levels of proBDNF in WT somas decreased to the same levels (WT = $0.847 ± 0.051$; DKO = $0.804 ± 0.038$; $p = 0.371$; Fig 6C). Conversely, addition of BDNF increased the levels of somatic proBDNF in WT but not in DKO neurons (WT = $1.071 ± 0.052$; DKO = $0.819 ± 0.05$; $p = 9.46 × 10^{-5}$; Fig 6C). This indicates that CaMKII is required for BDNF-induced (pro)BDNF expression.

To test whether BDNF-induced proBDNF expression involves the soluble N-ethylmaleimide-sensitive factor (NSF) attachment protein receptor (SNARE)-dependent BDNF secretion from the neurons themselves, cultures were infected with tetanus neurotoxin (TeNT), which cleaves vesicle-associated membrane protein (VAMP) 1–3 and blocks fusion of SVs and DCVs [59], at DIV 11 (Fig 6D). After 6 DIVs of TeNT expression, BDNF levels in synapses of WT neurons were reduced drastically (Fig 6E), comparable with untreated CaMKII DKO neurons (WT naive = $1 ± 0.083$; DKO Naive = $0.599 ± 0.035$; WT TeNT = $0.66 ± 0.05$; DKO TeNT = $0.71 ± 0.06$; Kruskal–Wallis $X^2$ [3,115] = 0.0002, Fig 6F). TeNT expression in CaMKII DKO cultures, on the other hand, did not alter the levels of synaptic BDNF. These data indicate that secreted BDNF requires CaMKII to increase neuromodulator expression.

## CaMKII is required for BDNF-induced BDNF accumulation via pCREB

In addition to activating CaMKII [56], BDNF is known to increase pCREB at S133 [60]. CaMKII regulates gene expression by a signal cascade that involves pCREB at S133 [27, 61, 62]. Thus, we tested the hypothesis that altering downstream targets of CaMKII in the signal cascade toward pCREB restores BDNF levels in neurons lacking CaMKII.

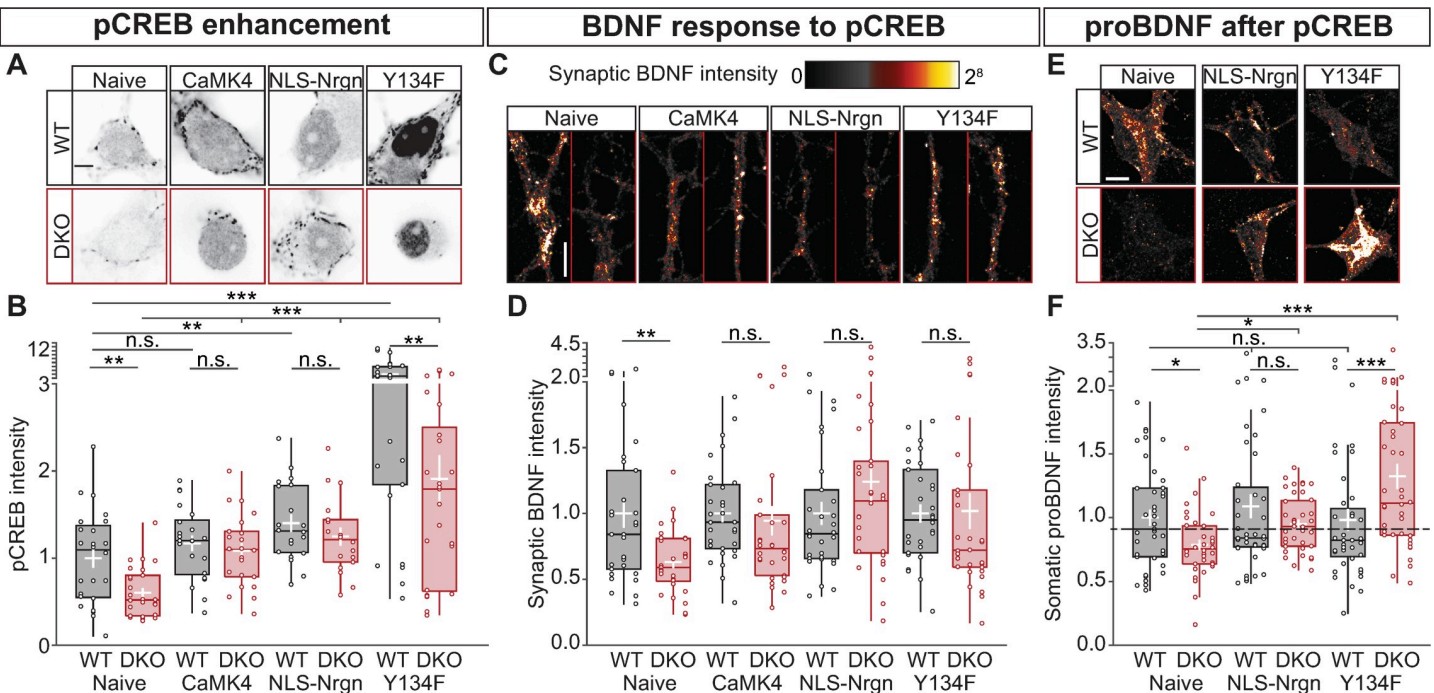

**Fig 7. CaMKII is required for BDNF-induced BDNF production via pCREB.** (A) Typical example of nuclei immunostained for pCREB at Ser-133. (B) Quantification of pCREB signal intensity normalized to WT (dashed line). (C) Typical examples of BDNF in synapses from untreated (Naive) cells, or treated with CaMKIV, NLS-Nrgn, or CREB-Y134F. (D) Quantification of synaptic BDNF intensity normalized to WT. (E) Typical examples of proBDNF in the soma from untreated neurons (Naive), treated NLS-Nrgn or CREB-Y134F. (F) Quantification of somatic proBDNF normalized to WT (dash line). Boxplots with 95% CI whiskers, white cross shows mean ± SEM. Columns and dots represents 0individual litters and neurons, respectively. The presented data can be found in S1 Data. $^*p < 0.05$, $^{**}p < 0.01$, $^{***}p < 0.001$. Scale bar = 20 μm (A-E) and 10 μm (C). BDNF, brain-derived neurotrophic factor; CaMKII, Ca2+/calmodulin-dependent kinase II; CaMKIV, Ca2+/calmodulin-dependent kinase IV; CREB, cAMP-response element binding protein; CREB-Y134F, phosphorylation of CREB with Tyr-to-Phe substitution at position 134; DKO, double-knockout; NLS-Nrgn, nuclear-localized neurogranin; n.s., not significant; pCREB, CREB phosphorylation; WT, wild type.

First, we compared pCREB levels in resting WT and CaMKII DKO neurons. Using a Ser133-specific antibody, we observed that pCREB levels were 40% lower in DKO neurons (WT = 1 ± 0.11; DKO = 0.6 ± 0.05; $p = 1.5 \times 10^{-3}$ Fig 7A and 7B). Second, we tested whether over-expression of 3 components in the signal cascade between CaMKII and pCREB restored a normal pCREB level in DKO neurons. The reduced pCREB levels were restored by addition of CaMKIV (WT = 1.169 ± 0.091; DKO = 1.082 ± 0.088; $p = 0.116$ Fig 7B), nuclear-localized neurogranin (NLS-Nrgn), which targets CaM to the nucleus and release it upon Ca$^{2+}$ influx [27] (WT = 1.4 ± 0.09; DKO = 1.24 ± 0.1; $p = 0.083$ Fig 7B), and a CREB mutation that lowers the threshold for S133 phosphorylation by PKA (Y134F) [63] (WT = 4.047 ± 0.568; DKO = 1.908 ± 0.271; $p = 0.002$ Fig 7B). Together, these data confirm that also in our system, CaMKII is required for normal pCREB levels and that overexpression of previously implicated components of the signal cascade between CaMKII and CREB restores a normal pCREB level in CaMKII DKO neurons.

We then tested if these components of the signal cascade between CaMKII and CREB also restore the reduced BDNF levels in CaMKII DKO synapses, using a phospho-dead mutant of CREB, with Ser-to-Ala substitution at position 133 (CREB-S133A) [64] as a negative control. Indeed, the lower synaptic BDNF staining was fully restored upon overexpression of CaMKIV, NLS-Nrgn, and phosphorylation of CREB with Tyr-to-Phe substitution at position 134 (CRE-B-Y134F) (naive: WT = 1 ± 0.1; DKO 0.63 ± 0.04, $p = 0.0078$; CaMKIV: WT = 1 ± 0.06; DKO 0.94 ± 0.11, $p = 0.077$; NLS-Nrgn: WT = 1 ± 0.08; DKO 1.24 ± 0.15, $p = 0.304$; CREB-Y134F: WT = 1 ± 0.08; DKO 1.018 ± 0.137, $p = 0.2$ Fig 7C and 7D). Expression of CREB-S133A led to reduced pCREB in unstimulated neurons, as expected [64] (S6 Fig), and BDNF levels were not

restored in CaMKII DKO synapses (WT = 1 ± 0.08; DKO = 0.52 ± 0.05; $p$ = 4.88 × 10$^{-5}$ S6 Fig). Together these data suggest that CaMKII regulates neuromodulation and synaptic BDNF levels via a signal cascade that involves calmodulin targeting to the nucleus, CaMKIV, and pCREB.

To confirm that the restored synaptic BDNF levels were due to new BDNF production, somatic proBDNF levels were also quantified after expression of NLS-Nrgn and CREBY134F (Fig 7E). The expression of either construct did not alter proBDNF levels in WT neurons. However, both restored the levels of proBDNF in DKO neurons (naive: WT = 1 ± 0.06, DKO 0.85 ± 0.05, $p$ = 0.0122; NLS-Nrgn: WT = 1.089 ± 0.093, DKO 0.951 ± 0.13, $p$ = 0.304; CRE-B-Y134F: WT = 1.982 ± 0.087, DKO 1.323 ± 0.098, $p$ = 2 × 10$^{-4}$ Fig 7F). Altogether, these data suggest that CaMKII is a critical factor in a positive feedback loop between secreted neuromodulators, such as BDNF, and the induction of the expression of their precursors via a CaMKII-mediated signal cascade that involves CaMKIV and pCREB.

### βCaMKII activity regulates neuropeptide levels

Because βCaMKII is known to be the dominant paralog to enhance pCREB [27] and βCaMKII, but not αCaMKII, rescued the total amount of CHGB and SCG2 (Fig 5D and 5E), we hypothesized that βCaMKII is necessary and sufficient for the regulation of neuropeptide expression and normal synaptic neuromodulator levels.

To test this hypothesis, we expressed α- and βCaMKII mutants that cannot be phosphorylated at their inhibitory sites, shown to cause a reduction in the threshold for LTP (long-active [LA], TT305/306VA for αCaMKII, and TT306/307VA for βCaMKII) [65, 66], or a kinase-dead ([KD] K42R for αCaMKII and K43R for βCaMKII) mutation [67] (Fig 8A) in CaMKII DKO neurons and immunostained them for the DCVs marker CHGB (Fig 8B) and endogenous BDNF (Fig 8C). Compared with mock infected DKO neurons, DKO neurons expressing βCaMKII-LA had increased levels of both CHGB (WT = 1 ± 0.08; DKO = 0.73 ± 0.05; α = 0.76 ± 0.1; αLA = 0.7 ± 0.07; αKD = 0.7 ± 0.08; β = 1.1 ± 0.15; βLA = 0.98 ± 0.13; βKD = 0.71 ± 0.07; Kruskal–Wallis X$^2$[7,231] = 0.0282 Fig 8B) and BDNF (WT = 1 ± 0.1; DKO = 0.52 ± 0.04; α = 0.52 ± 0.04; αLA = 0.57 ± 0.04; αKD = 0.63 ± 0.06; β = 0.63 ± 0.07; βLA = 0.84 ± 0.39; βKD = 0.6 ± 0.04; Kruskal–Wallis X$^2$(7,231) = 6.38 × 10$^{-5}$ Fig 8C). These levels were similar to WT cells. For CHGB, βCaMKII-WT rescued the immunoreactivity (Fig 8B), as suggested by the Western blot data (Fig 5D). The KD variant of βCaMKII did not rescue the reduced BDNF or CHGB immunoreactivity. These data confirm that βCaMKII has an important enzymatic function to regulate synaptic levels of neuropeptides and neurotrophic factors.

The dendritic length was also normalized to WT levels with overexpression of the constitutively active form of βCaMKII (S7 Fig), confirming its function in regulating dendritic arborization [41, 42], while synapse density was increased in all genotypes, except for the αCaMKII-LA (S7 Fig), indicating that αCaMKII activity is required to limit synaptogenesis in primary hippocampal neurons.

Together, these data indicate that the kinase activity of βCaMKII is critical to regulating neuropeptide levels at synapses. Thus, βCaMKII serves as a central link for BDNF-induced neuromodulator production controlling the signaling cascade toward pCREB, whereas it is dispensable for the trafficking of neuromodulator containing DCVs and their fusion at synapses (Fig 8D).

## Discussion

This study confirms that CaMKII regulates the secretion of neuromodulatory signals. However, the current data reveal that in mouse hippocampal neurons, the release machinery that

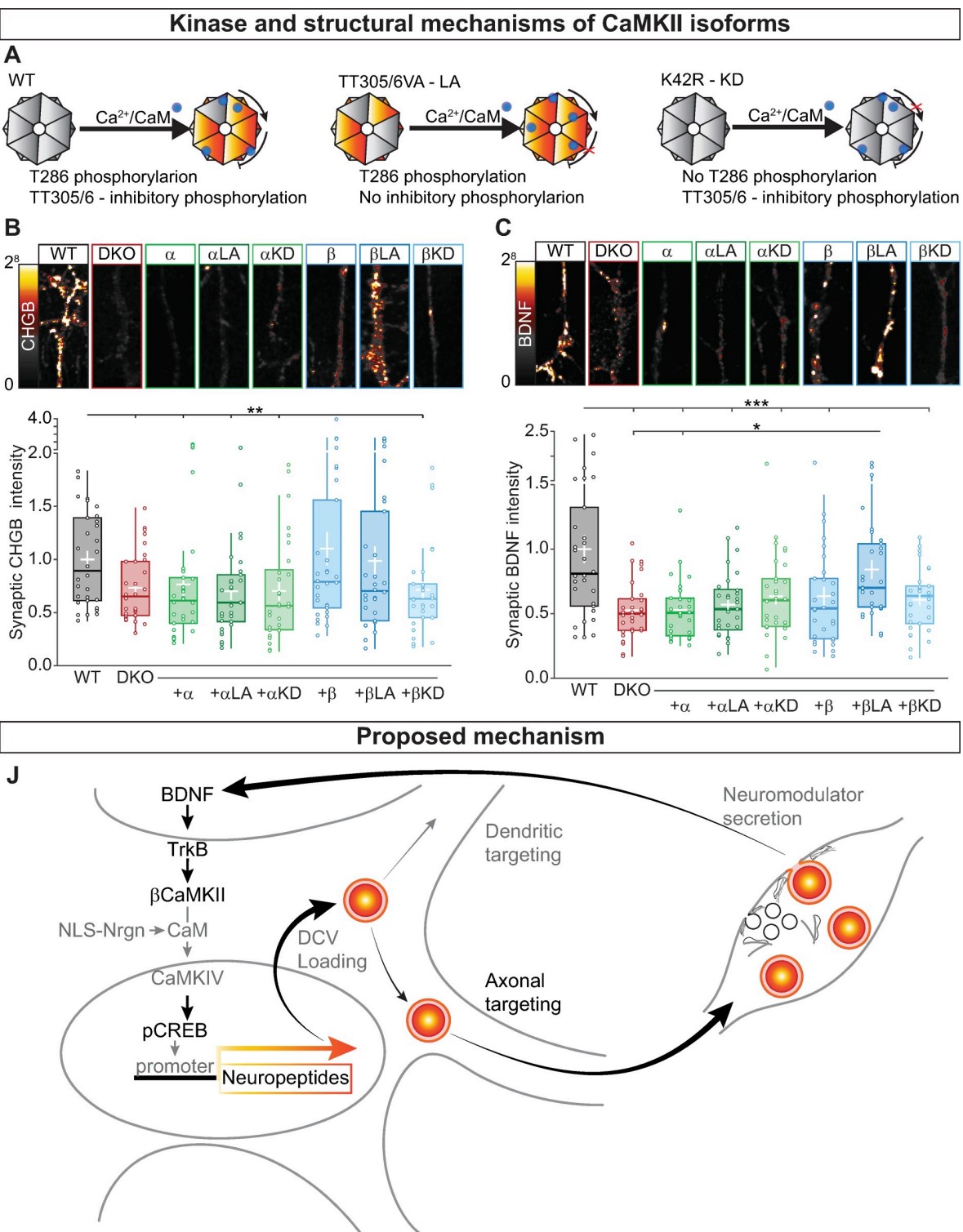

**Fig 8. CaMKII activity controls neuropeptide levels.** β (A) Schematic representation of WT αCaMKII (left) with Ca/CaM-induced autophosphorylation of T286 triggering the activity of the kinase. The TT305/306VA mutation (middle) inhibits TT305/6 inhibitory phosphorylation leading to an LA form of the kinase. The K42R mutation (right) does not allow T286 autophosphorylation producing a KD CaMKII. For βCaMKII, the mutations are TT306/307VA and K43R for the LA and KD, respectively. (B) Top, typical neurites immunostained for the endogenous DCV marker CHGB; bottom, quantification of synaptic CHGB intensity levels at VGLUT1 positive synapses for DKO and DKO rescued with the indicated constructs normalized to WT. (C) Top, typical neurites immunostained for

endogenous BDNF; bottom, quantification of synaptic BDNF intensity levels at VGLUT1 positive synapses for DKO and DKO rescued with the indicated constructs normalized to WT. (D) Schematic representation of the mechanism for CaMKII- and BDNF-dependent gene expression. BDNF binds to TrkB receptors; BDNF-TrkB binding leads to activation of βCaMKII that triggers a downstream pathway leading to Ca/CaM to enter into the nucleus and activating CaMKIV, which in turn phosphorylate CREB at S133. Neuropeptides and neuromodulator are then loaded into DCVs and targeted to dendrites and synapses, where they will fuse with the plasma membrane releasing their content. This mechanism will create a positive feedback loop to regulate neuromodulators content. Boxplots with 95% CI whiskers, white cross shows mean ± SEM. Columns and dots represents individual litters and neurons, respectively. The presented data can be found in S1 Data. *$p < 0.05$, **$p < 0.01$, ***$p < 0.001$. Scale bar = 10 μm (B and D). Ca/CaM, Ca2+/Calmodulin; CaMKII, Ca2+/calmodulin-dependent kinase II; CaMKIV, Ca2+/calmodulin-dependent kinase IV; CHGB, chromogranin B; CI, confidence interval; CREB, cAMP-response element binding protein; DCV, dense-core vesicle; DKO, double-knockout; KD, kinase-dead; LA, long-active; NLS-Nrgn, nuclear-localized neurogranin; SV, synaptic vesicle; Syn1, synapsin1; TrkB, tyrosine receptor kinase B; VAMP2, vesicle-associated membrane protein 2; VGLUT1, vesicular glutamate transporter 1; WT, wild type.

drives this process operates at a typical efficiency in the absence of α- and βCaMKII. Instead, we discovered that fewer DCVs that contain these neuromodulators were targeted into the axon, leading to fewer DCVs per synapse and less neuromodulator secretion during/after stimulation. This reduced axonal targeting was specific for DCVs; targeting of SV markers and exocytic proteins was only marginally reduced. The reduction of axonal targeting was explained mainly by a reduction in neuromodulator expression. We show that CaMKII is a crucial component in a positive feedback loop activated by secreted BDNF that promotes neuromodulator expression via βCaMKII, CaMKIV and pCREB. Finally, the kinase activity of βCaMKII, but not αCaMKII, was essential for normal neuromodulator expression and targeting to the axon.

CaMKII was already implicated in neuromodulator secretion based on pharmacological perturbations in mouse [10] and *Drosophila* [7, 8] and using genetic perturbation in *C. elegans* [9]. However, these studies reached different conclusions, based on different assays: pharmacological inhibition reduced the amount of DCV marker fluorescence lost from dendrites [10] or NMJ buttons [7] during stimulation, which was interpreted as reduced neuromodulator secretion. In contrast, mutant nematodes showed reduced DCV marker fluorescence in axons but a typical accumulation in coelomocytes, which was interpreted as abnormal DCV targeting but normal secretion [9]. Our study provides the first data that directly assess DCV fusion, at single vesicles' resolution, demonstrating that CaMKII is not required for the fusion process itself and thus agrees with the nematode study. Hence, pharmacological and genetic perturbations appear to result in different conclusions. The use of chemical inhibitors provides the most acute perturbations, but KN-93 is known to have side effects on the activity of other kinases and ion channel function [68].

One of the well-characterized actions of CaMKII is to couple excitation to transcription of neuronal plasticity genes, such as BDNF, via pCREB [25–27, 69]. Pharmacological inhibition of CaMKII represses the chromatin state of BDNF promoter [70] and blocks long-term depression (LTD)-mediated BDNF transcription [28]. Likewise, CHGB [48] and SCG2 [49] genes contain a CREB binding site, and their expression in PC12 cells is activity dependent [71]. In this study, we observed that the mRNA levels of CHGB, SCG2, and BDNF transcript II were also reduced, resulting in a reduced protein level of these DCV cargos, whereas a less activity-sensitive transcript of BDNF (transcript IV) [28, 30] was not reduced. Our data indicate that active βCaMKII expression restores CHGB and SCG2 protein levels (Fig 5D and 5E). βCaMKII selectively regulates excitation–translation coupling via CaMKIV-dependent pCREB [27] and, unlike αCaMKII, is expressed in both glutamatergic and GABAergic neurons [72]. Our study demonstrates that activating the signal cascade downstream of βCaMKII restores neuromodulator expression in a pCREB-dependent manner (Figs 7 and 8). Therefore, we conclude that βCaMKII activity, but not αCaMKII activity, induces expression of neuromodulators and other DVC cargo by activating a downstream pathway involving nuclear translocation of CaM, CaMKIV activation, and pCREB (Fig 8J).

Secreted neuromodulators like BDNF are known to activate CaMKII [56]. Our study suggests that this BDNF-induced CaMKII activity, especially βCaMKII activity, is part of a positive feedback loop to initiate a signal cascade that induces neuromodulator expression, e.g., to compensate for neuromodulators lost because of evoked secretion. Indeed, preventing BDNF secretion or scavenging of secreted BDNF in WT neurons produced a phenocopy of the situation in CaMKII DKO neurons (Fig 6). In addition, expression of components of the signal cascade downstream of βCaMKII activation restored the levels of BDNF in CaMKII DKO synapses (Fig 7). Thus, in addition with the well-established role of forming the holoenzyme together with αCaMKII and its targeting to the PSD [21, 22, 35, 69, 72], βCaMKII has a unique role not shared by αCaMKII. In mouse hippocampal neurons, βCaMKII acts in a positive feedback loop downstream of BDNF binding to Trk receptors and upstream of CaMKIV-dependent pCREB, which supports normal neuromodulator secretion (Fig 8).

Episomal expression of DCV cargo did not restore a normal amount of DCVs and BDNF at synapses. Hence, the availability of excess template is not sufficient to restore normal DCV/BDNF amounts. Although one such DCV cargo, e.g., BDNF/NPY, is overexpressed, expression of other DCV proteins that are also regulated by CaMKII, like CHGB, are most likely still reduced. This might prevent a normal DCV biogenesis rate, and the excess single overexpressed cargo might be subject to degradation.

Increasing pCREB restored normal levels of BDNF in DKO neurons but induced no further increase in WT neurons (Figs 6 and 7). This seems in contrast with previous observations that at least mRNA levels increase in WT neurons after BDNF application [53–55]. First, as in many other cases, mRNA levels may simply not predict protein levels, because of extensive posttranslational regulation. For instance, episomal expression of DCV cargo did not restore a normal amount of DCVs and BDNF at synapses, and such episomally encoded cargo might be subject to degradation (see previous) Second, the regulation of BDNF mRNA levels is complex and may differ under different experimental conditions. For instance, in immature cortical neuronal cultures, BDNF mRNA is differentially regulated via a CaMKII-dependent and a mitogen-activated protein kinase (MAPK)-dependent pathway [30, 54, 55]. The MAPK cascade, however, is not necessary to maintain normal BDNF mRNA transcription in unstimulated neurons [55].

Although the secretion of neuromodulatory signals is affected in the absence of α- and βCaMKII, secretion of SVs at single synapses was normal (S1A–S1F Fig) during high-frequency stimulation. These data are in line with previous studies in single null mutant neurons: at CA3 synapses, αCaMKII, or βCaMKII deficiency does not alter basal neurotransmission but abolishes LTP [21, 31]. In contrast, overexpression of α- or βCaMKII was shown to have opposite effects on spontaneous fusion of SVs in brain slices. The frequency of miniature excitatory postsynaptic current (mini-EPSC) was reduced in αCaMKII overexpression, whereas βCaMKII increased the frequency [72]. Such effects, mainly the increased frequency, may be explained by CaMKII's role in neuromodulation, as increased BDNF secretion was reported to increase mini-EPSC frequency [2]. Furthermore, we observed that dendritic and axonal length and synapse number were all increased in the absence of α- and βCaMKII (S3 Fig). Such changes in synapse number are also expected to affect overall spontaneous SV fusion frequency.

Both CaMKII isoforms are required for LTP in the hippocampus [21, 65, 73], and neuromodulators, especially BDNF, are too [74]. The 40% reduction in DCV fusion events per cell (Fig 1G) may help to link these 2 well-documented phenomena. However, the role of BDNF release in LTP is also demonstrated by acute pharmacological inhibition of CaMKII with KN-93, which strongly decreased the dendritic release of BDNF [10]. This acute effect is unlikely to

be explained by transcriptional regulation, as we report here for reduced BDNF release in our system. Hence, LTP probably involves additional KN-93 or CaMKII-dependent processes.

In *Drosophila* NMJ synapses, acute inhibition of CaMKII with KN-93 does not alter basal DCV fusion, but it limits the mobilization of stationary vesicles [33] and the capture at synapses of DCVs transiting along the axon [34]. This joint effect on mobilization and capture was proposed as a mechanism to support efficient DCV supply and fusion at NMJs during intense stimulation [7, 33]. Our data indicate that in mammalian neurons, the activity of CaMKII is dispensable for normal fusion efficiency. These mobilization and capture mechanisms may not be crucial in mammalian neurons, and they might be different between NMJ and CNS synapses or redundant mechanisms may have evolved to secure efficient fusion in the absence of α- and βCaMKII.

We observed that altered neuromodulation in the absence of CaMKII is explained by a robust and rather specific reduction of neuropeptides in synapses (Fig 3). A similar phenotype was observed in *C. elegans* motor neurons, where DCV cargo was also reduced in axons [9]. We observed that cargo loading in individual vesicles was unaltered (S1G–S1I Fig), indicating that the reduction of neuropeptide levels at synapses must be attributed to fewer DCVs and not too less cargo per vesicle, in line with observations in *C. elegans* motoneurons [9]. Interestingly, endogenous and exogenous DCV cargo was subject to the same reduction, suggesting that neuropeptide overexpression does not rescue the number of DCVs produced.

We observed that 50% fewer DCVs enter the axon of DKO neurons, whereas the same number entered into dendrites (Fig 4A–4G). This observation cannot be explained by an altered neuromodulator expression and probably involves CaMKII targets other than CREB. The targeting of vesicles to specific compartments involves post-translational modification of both cytoskeletal [75] and motor proteins [76]. For instance, a mutation in the CaMKII substrate cyclin-dependent kinase 5 [77] leads to the loss of neuropeptides targeting to the axon with a compensatory increase in dendritic targeting [78]. However, we did not observe such compensation, indicating that the altered targeting observed in the current study differs from what was previously reported. Therefore, CaMKII regulates neuromodulation by promoting DCV targeting to axons, with a mechanism that appears to be independent from dendritic targeting.

We also observed an increased number of retrograde moving vesicles in the axon (S3 Fig). This phenomenon could be a consequence of decreased synaptic capture of DCVs, which in *Drosophila* NMJ depends on CaMKII [8]. In NMJ boutons, electrical activity causes a rapid reduction of GFP-labeled neuropeptide signal [34]; this signal slowly recovers (25% recovery in 5 minutes) by capturing DCVs transiting along the axon in a process mediated by CaMKII activity [8]. However, we did not observe differences in the trafficking properties of DCVs in DKO axons, neither in baseline transport nor after high-frequency stimulation, where most DCVs reduce speed and arrest [46, 79]. In dendrites, CaMKII is required to disrupt the interaction between KIF17 and Mint1, allowing cargo unloading from the motor complex [23], and calmodulin was recently reported to regulate DCV-kinesin loading and unloading in dendritic spines [80]. These considerations strengthen the idea that in addition to regulating neuromodulator expression, CaMKII also regulates DCV targeting to axons.

In conclusion, the current study provides direct evidence that CaMKII is dispensable for efficient neuromodulator exocytosis in mouse hippocampal neurons and that βCaMKII regulates neuromodulation by promoting CaMKIV-dependent neuropeptide expression downstream of BDNF signaling. Hence, CaMKII is crucial in a feedback loop coupling neuromodulator secretion to neuromodulators expression and subsequent DCV targeting into the axon.

## Materials and methods

### Ethical statement

All animals were housed and bred according to institutional and Dutch Animal Ethical Committee regulations (DEC-FGA 11–03).

### Laboratory animals and primary neuron cultures

The generation of floxed *Camk2a* and *Camk2b* mice was previously described (Kool 2019). Postnatal day 1 (P1) pups were humanely killed, and hippocampi were dissected in Hank's balanced salt solution (HBSS Sigma) with 10 mM HEPES (Life Technology), digested in 0.025% trypsin (Life Technologies) for 20 minutes at 37°C and dissociated with fire-polished Pasteur pipettes. Dissociated neurons were resuspended in neurobasal supplemented with 2% B-27, 18 mM HEPES, 0.25% Glutamax, 0.1% penicillin/streptomycin (Life Technologies) and plated at a density of 1,300 neurons/well on astrocyte micro-island [6, 81, 82] in 12-well plates; for high-density cultures 25,000 neurons/well were plated on pregrown glia cells. Astrocyte micro-island were generated by plating 6,000 rat glial cells per agarose-coated 18-mm glass coverslip, stamped with 0.1 mg/ml poly-D-lysine (Sigma) and 0.7 mg/ml rat tail collagen (BD Biosciences). For Western blots and qRT-PCR, neurons were plated at a density of 250,000 neurons/well on plates coated with a solution of 0.0005% Poly-L-ornithine, 2.5 µg/ml Laminin (Sigma). Neuronal cultures were kept in supplemented neurobasal at 37°C and 5% $CO_2$.

### Plasmid and lentiviral infection

pCMV(pr)Camk2a-pCamkII(pr)eCFP and pCMV(pr)Camk2b-pCamkII(pr)eCFP were created by substituting tdTomato to eCFP from the plasmid in [83]. pSyn(pr)Synapsin-mCherry [39], pSyn(pr)BDNF-pHluorin, and pSyn(pr)BDNF-mCherry were engineered by replacing NPY with cDNA of BDNF from the plasmid encoding pSyn(pr)NPY-pHluorin [38] and pSyn(pr)NPY-mCherry [38]. pSyn(pr)NaV$_{II-III}$-BFP was created substituting BFP from YFP in NaV$_{II-III}$-YFP [44, 82]. A TetON expression system containing NPY-mCherry and mGFP was used for the trafficking experiments and to determine the axonal fusion of DCV [6]. pCMV(pr)HA-NLS-Nrgn was produced as described in (Ma and colleagues 2014); pCMV(pr)HA-CaMKIV, pCMV(pr)HA-CREB-S133A, and pCMV(pr)HA-CREB-Y134F were cloned from a mouse cDNA library.

All plasmids were sequenced-verified and subcloned into lentiviral vectors to produce viral particles [84].

Neuronal cultures were infected with Cre-recombinase 5 hours after plating to generate CaMK2-DKO or an inactive Cre [36] as control. Rescue plasmids were added to the same mix. For experiments in Fig 1 and Fig 2, viruses encoding the BDNF-pHluorin and Synapsin-mRFP were added to the neuronal culture 6 days before the readout. For experiments in Fig 4A–4G, viruses encoding BDNF-mCherry and Nav$_{II-III}$-YFP were added to the neuronal culture 3 days prior to live-cell imaging; experiments in Fig 4H–4N are described next for sparse labeling. For experiments in Fig 7A–7C, 0.2 µg/ml of BDNF antibody (DSHB) or 100 ng/ml or recombinant BDNF (Bio Connect) were added twice to the culture every 24 hours from DIV15, and cells were fixed 48 hours after the first administration. For experiments in Fig 7D–7F, viruses encoding for TeNT-IRES2-mCherry were added 6 days before fixation. For experiments in Fig 8, viruses encoding HA-CaMKIV, HA-NLS-Nrgn, and HA-CREB-Y134F were added to the culture 48 hours before fixation. Proper concentration of viruses was tested in WT cultures.

For sparse labeling with lentivirus particles, neurons were incubated in supplemented neurobasal with adequate amount of viral particles for 2 hours at 37°C, 5% $CO_2$, washed in Dulbecco's modified Eagle medium (DMEM Life Technologies) containing 10% fetal calf serum (FCS), resuspended in supplemented neurobasal and plated 5,000 neurons/well on top of previously plated high-density culture, resulting in a 30,000 neuron/well cultures with 15% infected neurons/well. Expression was started 6 days before experiment by adding 2 g/ml doxycycline hyclate (Sigma).

## Live-cell imaging

Neurons at DIV 17–18 were placed in the imaging chamber containing Tyrode's solution (2 mM $CaCl_2$, 2.5 mM KCl, 119 mM NaCl, 2 mM $MgCl_2$, 30 mM glucose, 25 mM HEPES [pH 7.4]). Experiments were performed at room temperature (RT) with superfusion of Tyrode's buffer unless otherwise specified. Images were acquired on an Axiovert II microscope (Zeiss, Oberkochen, Germany) with a 40× oil objective (NA = 1.3) for SV fusion, DCV fusion, and trafficking, or on a Nikon Eclipse Ti microscope with 63× oil objective (NA = 1.4) for axonal targeting. Time-lapse recordings were acquired using Metamorph 6 (Molecular Devices https://www.moleculardevices.com) and an EM-CCD camera or using NisElements 4.30 software (NIKON, https://www.microscope.healthcare.nikon.com). The acquisition frequency was 2 Hz for pHluorin-based assays and 1 Hz for trafficking and targeting assays.

For SypHy experiments, the imaging protocol included 30 seconds of baseline recording, electrical field stimulation using a A-385 stimulus isolator (WPI) controlled by a Master 8 (AMPI), delivering 1-millisecond, 30-mA pulses for 5 seconds at 40 Hz, followed by 1 minute of recovery time and a final 5 seconds' perfusion with modified Tyrode's containing $NH_4Cl$ (2 mM $CaCl_2$, 2.5 mM KCl, 119 mM NaCl, 2 mM $MgCl_2$, 30 mM glucose, 25 mM HEPES, 50 mM $NH_4Cl$ [pH 7.4]) delivered by gravity flow through a capillary placed above the cell.

For DCV fusion, neurons were imaged for 30 seconds as baseline, stimulated with electrical field stimulation for 8 pulses of 1 second at 50 Hz separated by 0.5 seconds, allowed to rest for 30 seconds after which the stimulation was repeated. After an additional 30 seconds of recovery Tyrode's containing $NH_4Cl$ were superfused for 5 seconds.

For trafficking experiments, labeled neurons were identified, and the field of view was adjusted to fit the majority of the axon. After 30 seconds of baseline, electrical field stimulation was applied with 16 bursts of 1 second at 50 Hz separated by 0.5 seconds. For fluorescence recovery after photobleaching (FRAP) experiments, laser intensity and pulse duration for bleaching were optimized to reach >90% fluorescence decrease of mCherry [82], images were acquired for 180 seconds after an initial 5 seconds of baseline.

## Immunostaining and confocal microscopy

Neurons were fixed at DIV18 with 3.7% formaldehyde (Merck) in phosphate-buffered saline ([PBS] 137mM NaCl, 2.7 mM KCl, 10 mM Na2HPO4, 1.8 mM KH2PO4 [pH 7.4]) for 20 minutes. Cells were permeabilized in 0.5% TritonX-100 (Fisher Chemical) for 5 minutes and blocked with 0.1% TritonX-100 and 2% normal goat serum for 60 minutes. Primary antibody incubation with MAP2 (Abcam, 1:1,000), SMI312 (Biolegend, 1:1,000), CaMK2 (SantaCruz, 1:350), VGLUT1 (Millipore, 1:5,000), BDNF (DSHB, 1:10) proBDNF(Alomene Labs, 1:300), CHGB (SySy, 1:500), SCG2 (Biodesign International, 1:1,000), synaptophysin 1 (SySy; 1:1,000), VAMP2 (SySy, 1:1,000), synaptotagmin 1 (W855 a kind gift from T. Südhof, Stanford, CA; 1:2000), MUNC18-1 (BD, 1:1,000), SNAP-25 (Abcam, 1:1,000), phosphoCREB-S133 (Abcam, 1:200) was performed for 2 hours at RT. Alexa Fluor conjugated secondary antibodies (1:1,000; Invitrogen) were incubated for 1 hour at RT. Coverslips were mounted in Mowiol

and imaged on Nikon Eclipse Ti microscope confocal laser-scanning microscope (40× objective; NA 1.3) with NisElements 4.30 software. Images were acquired as Z-stack of 3 planes 250 μm apart and 4 images per plane with 15% overlap to ensure that the entire micro-island was in the field of view.

## Western blot

Hippocampal neurons were lysed at DIV18. Lysates were run on an SDS-PAGE and transferred to a polyvinylideenfluoride (PVDF) membrane (Bio-Rad). Membranes were blocked with 5% milk (Merk) in PBS with 0.1% Tween-20 (PBST) for 1 hour at RT; incubated in primary antibody against CaMK2 (Transduction lab, 1:2,500), CHGB (SySy, 1:1,000), SCG2 (GeneTex–bioconnect, 1:1,000), actin (Chemicon, 1:10,000) overnight at 4°C. Secondary alkaline phosphatase conjugated antibodies (Jackson Immuno Research, 1:10,000) were incubated for 30 minutes at RT. Membranes were visualized with AttoPhos (Promega) and scanned with a FLA-5,000 fluorescent image analyzer (Fujifilm).

## Quantitative RT-PCR

Total RNA was extracted from whole brains using UltraClean Tissue & Cells RNA Isolation kit (15000–50; MO BIO, Carlsbad, CA, USA). Synthesis of cDNA was performed using oligo d (T) and random hexamers with the kit iScriptTM select cDNA Synthesis Kit (1708896; BIO-RAD, Madrid, Spain). Quantitative RT-PCR was performed with SensiFastTM SYBR Lo-Rox kit (BIO-94005; BIOLINE, London, UK) using de Light Cycler 480 System (Roche Applied Science, Woerden, The Netherlands) with the following primers: SCG1-F:GTCCTC TCAAATGCCCTATCCA; SCG1-R:ACTTCGAGTTCTGGTTTTCACC; SCG2-F:GCTGTC CGGTGCTGAAA; SCG2-R:TTAGCTCCAGCCATGTCTTAAA; BDNF transcript2-F:CCAT CCACACGTGACAAAAC; BDNF transcript2-R:GGTGCTGAATGGACTCTGCT; BDNF transcript4-F:GACCAGAAGCGTGACAACAA; BDNF transcript4-R:AGGGTCCACACA AAGCTCTC; eEF2a-F:CAATGGCAAAATCTCACTGC; and eEF2a-R:AACCTCATCTC TATTAAAAACACCAAA.

eEF2a was used as reference gene for normalizing the data across samples. The 2ΔΔCt method was used for calculating the fold change in expression of the different genes.

## Image analysis

For SV fusion analysis, individual synapses were identified and treated as single region of interest (ROI) with a custom-made ImageJ algorithm (National Institute of Health); briefly, synapses were identify based on their increase in signal during $NH_4$ application, if upon $NH_4$ an ROI had a $\Delta F/F_0 < 4 \,{}^* \mathrm{StD}(F_0) + F_0$, where StD represents the standard deviation of the signal, the ROI was discarded. Individual traces were analyzed with a custom-made MATLAB (Mathworks, www.mathworks.com) script where synapses were quantified as active if the maximum $\Delta F/F_0$ upon stimulation was $\geq 3 \,{}^* \mathrm{StD}(F_0)$; active synapses were pooled per cell. SypHy fusion fraction was calculated as the $\Delta F_{\mathrm{stimulation}} / \Delta F_{\mathrm{NH4}}$.

For DCV fusion, 2×2 pixel ROIs were placed semi-automatically, using a custom-made script in ImageJ, on all pHluorin region that appeared during the electrical stimulation. Individual traces were validated using a custom-made MATLAB script; only regions that showed an increase in $\Delta F/F_0 > 3 \,{}^* \mathrm{StD}(F_0) + F_0$ and with a rise time <1 seconds were considered as positive fusion events. The total number of DCVs was calculated based on the $NH_4Cl$ response of individual recording with a custom-made ImageJ algorithm. Because of the overlap of DCVs in individual puncta, the number of vesicles was corrected normalizing the puncta intensity by the mode of the first percentile of the intensity distribution per cell.

For neuronal morphology and synaptic quantification, maximum intensity projections of confocal images were analyzed with a custom-made ImageJ algorithm, dendrites and axons based on Ridge detection, and their length was calculated based on the skeleton analysis in ImageJ; synapses were identified based on their intensity and dimension. For signal intensity, the intensity of each individual neuron was normalized to the average intensity of the WT condition in that biological replica.

For Western blot analysis, the band at the proper molecular weight was considered as an ROI. Intensity density was used to normalize the signal using the density of actin as reference. The adjusted signal was normalized to the signal of the WT condition.

For visualization purposes, brightness and contrast of representative examples were adjusted in a linear scale using the WT as reference. Saturation was always <1% of the pixels and 0% of the pixels were set to undersaturation during adjustment.

## Statistics

Anderson–Darling test was used to test normality distribution of the data, and Levene's test for homogeneity of variances. When the assumptions for parametric tests were met, $t$ test and 1-way ANOVA (post hoc Dunn–Sidak or Fisher) were used to test significant differences in the mean of the population. In case data were nonparametric, Mann–Whitney U and Kruskal–Wallis tests were used to test the significance of the median of the populations. All statistics were done in MATLAB, and their values are reported in S1 Data.

## Supporting information

**S1 Movie. DCV fusion observed with BDNF-pHluorin.** BDNF, brain-derived neurotrophic factor; DCV, dense-core vesicle.
(MP4)

**S2 Movie. BDNF containing vesicles trafficking at the AIS.** AIS, axon initial segment; BDNF, brain-derived neurotrophic factor.
(MP4)

**S3 Movie. Axonal DCV trafficking observed with NPY-mCherry.** DCV, dense-core vesicle; NPY, neuropeptide Y.
(MP4)

**S1 Fig. CaMKII does not affect single vesicles' properties.** (A) Representative images of WT and DKO neurons infected with SypHy at the end of the electrical stimulation (left) and kymograph showing the dynamics of SypHy signal over time. (B) Average SypHy traces normalized as $\Delta F/F_0$. After 5 seconds of 40-Hz stimulation (light blue bar), neurons were allowed to rest for 60 seconds before $NH_4Cl$ superfusion to reveal the total amount of SypHy per synapses (light green bar). (C) Percentage of active synapses during a 40-Hz, 5-second stimulation. (D) SypHy fused fraction at the end of 5 seconds of 40-Hz stimulation. (E) Quantification of the decay constant τ for the SypHy signal intensity decay after stimulation. (F) Quantification of the SypHy signal upon $NH_4^+$ superfusion. (G) Typical neurite expressing BDNF-pHluorin during baseline, b, and during $NH_4^+$ superfusion, n. (H) Quantification of BDNF-pHluorin baseline fluorescence before stimulation. (I) Probability distribution of cells containing determined amount of BDNF-pHluorin positive puncta. (L) Average traces of BDNF-pHluorin fusion events aligned at the moment of fusion (0 seconds). (M) Histogram showing the BDNF-pHluorin signal intensity of individual fusion events. The fusion intensity was calculated as the fold change in fluorescence intensity from the 5 frames before fusion to the maximum intensity during fusion. (N) Quantification of average BDNF-pHluorin fusion intensity

per cell. Traces show mean ± SEM (shaded area), boxplots with 95% CI whiskers, central bar is the median, white cross shows mean ± SEM. Columns and dots represent individual litters and neurons, respectively. The presented data can be found in S1 Data. *$p < 0.05$, **$p < 0.01$, ***$p < 0.001$. Scale bar = 25 μm (A images), 50 μm (A kymograph); kymograph color map in 1A, NanoJ-Orange in ImageJ. BDNF, brain-derived neurotrophic factor; CaMK, Ca2+/calmodulin-dependent kinase II; CI, confidence interval; SypHy, synaptophysin-pHluorinl; WT, wild type.
(TIF)

**S2 Fig. CaMKII does not regulate expression of plasmids driven by Synapsin promoter.**
(A) Typical images of neurites of WT (left) and DKO (right) immunostained for BDNF. Right, quantification of BDNF intensity at VGLUT1 labeled synapses in WT and DKO neurons. (B) Typical images of neurites of WT (left) and DKO (right) immunostained for CHGB. Right, quantification of CHGB intensity at VGLUT1 labeled synapses in WT and DKO neurons. (C) Typical images of neurites of WT (left) and DKO (right) expressing NPY-pHluorin. Right, quantification of NPY-pHluorin intensity at VGLUT1-labeled synapses in WT and DKO neurons. (D) Cumulative probability and average intensity for single BDNF puncta. (E) Cumulative probability and average intensity for single CHGB puncta. (F) Cumulative probability and average intensity for single NPY-pHluorin puncta. (G) Typical neurons overexpressing Synapsin-mCherry (H) Quantification of the average intensity of Synapsin-mCherry per cell. (I) Typical neuron overexpressing mGFP. (J) Quantification of average mGFP intensity in the total neuritic arbor. Boxplots with 95% CI whiskers, white cross shows mean ± SEM, central bar is the median. Columns and dots represent individual litters and neurons, respectively. The presented data can be found in S1 Data. *$p < 0.05$, **$p < 0.01$, ***$p < 0.001$. Scale bar = 25 μm (G-I). BDNF, brain-derived neurotrophic factor; CaMKII, Ca2+/calmodulin-dependent kinase II; CHGB, chromogranin B; CI, confidence interval; DKO, double-knock-out; mGFP, membrane-bound GFP; NPY, neuropeptide Y; SypHy, synaptophysin-pHluorin; VGLUT, vesicular glutamate transporter; WT, wild type.
(TIF)

**S3 Fig. Neuromodulators levels are not dependent on their distance from the soma.** (A) Typical images of neurites of WT (top) and DKO (bottom) immunostained for BDNF. Right, quantification of BDNF intensity at the soma of WT and DKO neurons. (B) Typical synapses immunostained for BDNF at the indicated distances from the soma. (C) Intensity profile of BDNF immunoreactivity in correlation to the distance from the soma. (D) Intensity profile of BDNF in VGLUT1 positive synapses in correlation to the distance from the soma. (E) Typical images of neurites of WT (top) and DKO (bottom) immunostained for CHGB. Right, quantification of CHGB intensity at the soma of WT and DKO neurons. (F) Typical synapses immunostained for CHGB at the indicated distances from the soma. (G) Intensity profile of CHGB immunoreactivity in correlation to the distance from the soma. (H) Intensity profile of CHGB in VGLUT1 positive synapses in correlation to the distance from the soma. (I) Typical images of neurites of WT (top) and DKO (bottom) expressing NPY-pHluorin. Right, quantification of NPY-pHluorin intensity at the soma of WT and DKO neurons. (J) Typical synapses expressing NPY-pHluorin at the indicated distances from the soma. (K) Intensity profile of NPY-pHluorin immunoreactivity in correlation to the distance from the soma. (L) Intensity profile of NPY-pHluorin in VGLUT1 positive synapses in correlation to the distance from the soma. Boxplots with 95% CI whiskers, white cross shows mean ± SEM, central bar is the median. Columns and dots represent individual litters and neurons, respectively. The presented data can be found in S1 Data. *$p < 0.05$, **$p < 0.01$, ***$p < 0.001$. Full figure width = 36.46 μm (A-E-I) 4.25 μm (B-F-J). BDNF, brain-derived neurotrophic factor; CHGB, chromogranin B;

CI, confidence interval; DKO, double-knockout; mGFP, membrane-bound GFP; NPY, neuro-peptide Y; SypHy, synaptophysin-pHluorin; VGLUT, vesicular glutamate transporter; WT, wild type.
(TIF)

**S4 Fig. CaMKII negatively regulates neurites length and synaptogenesis.** (A) Typical neu-rons grown on astrocyte micro-island immunostained for the dendritic marker MAP2 (green) and the synaptic vesicle marker VGLUT1 (magenta). In the black and white bottom insert, the same neurons immunostained for CaMKII. (B) Quantification of average somatic CaMKII intensity. (C) Quantification of average dendritic length. (D) Quantification of synapse density (number of synapses per mm of dendrite). (E) Sholl analysis for the distribution of dendrite crossings. (F) Typical neurons grown on astrocytes micro-islands immunostained for the axo-nal marker SMI312 and synaptic vesicle marker VGLUT1. In zooms (bottom), light green lines define axons, and the VGLUT1 signal is inverted. (B) Quantification of average axonal length per cell in mm. (C) Quantification of synapse density (number of synapses per mm of axon). (D) Sholl analysis of synapse localization. (F) Sholl analysis for the distribution of axo-nal crossings. Traces shows mean ± SEM (shaded area), boxplots with 95% CI whiskers, white cross shows mean ± SEM. Columns and dots represent individual litters and neurons, respec-tively. The presented data can be found in S1 Data. $^*p < 0.05$, $^{**}p < 0.01$, $^{***}p < 0.001$. Scale bar = 50 μm (A). CaMKII, Ca2+/calmodulin-dependent kinase II; MAP2, microtubule-associ-ated protein 2; VGLUT, vesicular glutamate transporter; WT, wild type.
(TIF)

**S5 Fig. CaMKII does regulates the amount of BDNF vesicles entering the soma, not DCV trafficking.** (A) Schematic representation of AIS targeting; the region of the AIS was visualized by Na$_v$II-III BFP. BDNF-mCherry vesicles were bleached at the AIS to allow quantification of new vesicles entering the axon. (B) Quantification of DCV flux at the AIS calculated as the number of BDNF-mCherry positive puncta that enter the Na$_v$II-III BFP area per minute in ret-rograde (from the axon to the soma) direction. (C) Quantification of BDNF-mCherry speed at the AIS in the retrograde direction. (D) Quantification of the run length in μm at the AIS in the anterograde (left) and retrograde (right) direction. (E) Quantification of the pausing time, as percentace of the total moving time, at the AIS in the anterograde (left) and retrograde (right) direction. (F) Schematic representation of the TrkB endocytosis assay. Neurons were incubated with TrkB antibody for 30 minutes at 37˚C. After fixation and prior to permeabiliza-tion, cultures were incubated with saturating concentration of secondary antibody. Endocy-tosed TrkB was visualized using standard immunostaining procedures. Bottom, typical WT neurites immunostained for MAP2 (green) and endocytosed TrkB (magenta). (G) Quantifica-tion of the average number of TrkB puncta per mm of MAP2 positive neurites. (H) Quantifica-tion of the average intensity of TrkB puncta. (I) Schematic representation of the surface TrfR assay. Neurons were incubated with Trf-Alexa-488 (Trf-488) antibody for 5 minutes after blocking endocytosis. Total amount of TrfR was visualized using standard immunostaining procedures. Bottom, typical WT neurites immunostained for Trf-488 (green) and TrfR (magenta). (J) Quantification of the total TrfR levels. (K) Quantification of the ratio between Trf-488 intensity to TrfR intensity. (L) Schematic representation of NPY-mCherry puncta moving in the anterograde and the retrograde direction in kymographs, light blue bars repre-sent the stimulation of 16 trains of 50 APs at 50 Hz. Trafficking parameters were calculated before stimulation (B), during stimulation (S) and after stimulation (R). (M) Quantification of the axonal density of DCV moving in the anterograde and retrograde direction. (N) Quantifi-cation of the average speed (in μm/s) for NPY-mCherry puncta, per cell, in the anterograde and retrograde direction. (O) Quantification of the run length (in μm) for NPY-mCherry

puncta, per cell, in the anterograde and retrograde direction. (P) Quantification of the percentage of time DCVs pause in the anterograde and retrograde direction. Traces shows mean ± SEM (shaded area), boxplots with 95% CI whiskers, white cross shows mean ± SEM, central bar is the median. Columns and dots represent individual litters and neurons, respectively. The presented data can be found in S1 Data. $^*p < 0.05$, $^{**}p < 0.01$, $^{***}p < 0.001$. Scale bar = 25 μm (F-I). AIS, axon initial segment; BDNF, brain-derived neurotrophic factor; BFP, blue fluorescence protin; CaMKII, Ca2+/calmodulin-dependent kinase II; CHGB, chromogranin B; CI, confidence interval; DCV, dense-core vesicle; mGFP, membrane-bound GFP; Nav$_{II-III}$, voltage-gated sodium channel intracellular domain; NPY, neuropeptide Y;; TrfR, transferrin receptor; TrkB, tyrosine receptor B; VGLUT, vesicular glutamate transporter; WT, wild type.
(TIF)

**S6 Fig. CREB phosphorylation at Ser-133 is required for BDNF accumulation in synapses.** (A) Typical example of the downstream pathway of βCaMKII that lead to pCREB at Ser-133. CREB phosphorylation can be enhanced in the absence of CaMKII by shuttling CaM via NLS-Nrgn, by activating CaMKIV or by lowering the threshold for PKA-mediated CREB-Y134F. (B) Percentage of cells that presented pCREB in unstimulated conditions. (C) Quantification of the relative intensity of BDNF in synapses of WT and DKO neurons left untreated or with expression of the phosphodead form of CREB (CREB-S133A). Boxplots with 95 CI whiskers, white cross shows mean ± SEM. Columns and dots represent individual litters and neurons, respectively. The presented data can be found in S1 Data. $^*p < 0.05$, $^{**}p < 0.01$, $^{***}p < 0.001$. BDNF, brain-derived neurotrophic factor; CaM, calmodulin; CaMKII, Ca2+/calmodulin-dependent kinase II; CaMKIV, Ca2+/calmodulin-dependent kinase IV; CI, confidence interval; CREB, cAMP-response element binding protein; CREB-Y134F, phosphorylation of CREB with Tyr-to-Phe substitution at position 134; DKO, double-knockout; NLS-Nrgn, nuclear-localized neurogranin; pCREB, CREB phosphorylation; PKA, protein kinase A; WT, wild type.
(TIF)

**S7 Fig. α and βCaMKII have a different function in neuronal morphology.** (A) Typical neurons grown on astrocyte micro-islands immunostained for the dendritic marker MAP2 (green) and the SV marker VGLUT1 (magenta). (B) Average dendritic length in mm. (C) Synapse distribution expressed in the number of synapses per mm of dendrite. (D) Average axonal length in mm. Boxplots with 95 CI whiskers, white cross shows mean ± SEM. Columns and dots represent individual litters and neurons, respectively. The presented data can be found in S1 Data. $^*p < 0.05$, $^{**}p < 0.01$, $^{***}p < 0.001$. Scale bar = 50 μm (A). CaMKII, Ca2+/calmodulin-dependent kinase II; CI, confidence interval; MAP2, microtubule associated protein 2; SV, synaptic vesicle; VGLUT1, vesicular glutamate transporter 1.
(TIF)

**S1 Data. Source data for each figure.**
(XLSX)

**S1 Raw Images. Original immunoblot gel for Fig 1 and Fig 5.**
(PDF)

## Acknowledgments

The authors thank Joke Wortel for animal breeding, Joost Hoetjes for genotyping, Robbert Zalm for cloning and producing viral particles, Desiree Schut for astrocytes culture and cell

culture assistance, Frank den Oudsten for astrocytes culture and qRT-PCR assistance, Ingrid Saarloos for assistance in protein chemistry and Western blot, Aygul Subkhangulova, members of the CNCR DCV project team, and Ype Elgersma (Erasmus MC, Rotterdam) for fruitful discussions.

## Author Contributions

**Conceptualization:** Alessandro Moro, Ruud F. Toonen, Matthijs Verhage.

**Data curation:** Alessandro Moro, Ruud F. Toonen, Matthijs Verhage.

**Formal analysis:** Alessandro Moro.

**Funding acquisition:** Matthijs Verhage.

**Investigation:** Alessandro Moro.

**Methodology:** Alessandro Moro, Ruud F. Toonen.

**Project administration:** Matthijs Verhage.

**Resources:** Geeske M. van Woerden.

**Software:** Alessandro Moro.

**Supervision:** Ruud F. Toonen, Matthijs Verhage.

**Validation:** Alessandro Moro.

**Visualization:** Alessandro Moro, Ruud F. Toonen, Matthijs Verhage.

**Writing – original draft:** Alessandro Moro, Geeske M. van Woerden, Ruud F. Toonen, Matthijs Verhage.

**Writing – review & editing:** Alessandro Moro, Geeske M. van Woerden, Ruud F. Toonen, Matthijs Verhage.

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
