## [Editor Report · Decision Letter 0]

6 Oct 2019

Dear Dr Verhage, 

Thank you for submitting your manuscript entitled "CaMKII controls neuromodulation via neuropeptide gene expression and axonal targeting of neuropeptide vesicles." for consideration as a Research Article by PLOS Biology.

Your manuscript has now been evaluated by the PLOS Biology editorial staff as well as by an academic editor with relevant expertise and I am writing to let you know that we would like to send your submission out for external peer review.

Please re-submit your manuscript within two working days, i.e. by Oct 08 2019 11:59PM.

Kind regards,

Ines

--

Ines Alvarez-Garcia, PhD

Senior Editor

PLOS Biology

Carlyle House, Carlyle Road

Cambridge, CB4 3DN

+44 1223–442810

---

## [Decision Letter · Decision Letter 1]

12 Nov 2019

Dear Dr Verhage,

Thank you very much for submitting your manuscript "CaMKII controls neuromodulation via neuropeptide gene expression and axonal targeting of neuropeptide vesicles" for consideration as a Research Article at PLOS Biology. Your manuscript has been evaluated by the PLOS Biology editors, an Academic Editor with relevant expertise, and by three independent reviewers.

The reviews of your manuscript are appended below. As you will see, the reviewers find the work potentially interesting, however they also feel that additional mechanistic insights would be required for this work to be appropriate for PLOS Biology. They would like you to focus on two aspects: 1) mechanisms on how CaMKII beta regulates BDNF transcription, and 2) how these findings relate to synapse development and plasticity.

Following a discussion with the Academic Editor, I regret that we cannot accept the current version of the manuscript for publication. We remain, however, interested in your study and we would be willing to consider resubmission of a comprehensively revised version that thoroughly addresses all the reviewers' comments. We cannot make any decision about publication until we have seen the revised manuscript and your response to the reviewers' comments. Your revised manuscript would be sent for further evaluation by the reviewers.

We appreciate that these requests represent a great deal of extra work, and we are willing to relax our standard revision time to allow you [six months] to revise your manuscript. Please email us (plosbiology@plos.org) to discuss this if you have any questions or concerns, or think that you would need longer than this. At this stage, your manuscript remains formally under active consideration at our journal; please notify us by email if you do not wish to submit a revision and instead wish to pursue publication elsewhere, so that we may end consideration of the manuscript at PLOS Biology.

Your revisions should address the specific points made by each reviewer. Please submit a file detailing your responses to the editorial requests and a point-by-point response to all of the reviewers' comments that indicates the changes you have made to the manuscript. In addition to a clean copy of the manuscript, please upload a 'track-changes' version of your manuscript that specifies the edits made. This should be uploaded as a "Related" file type. You should also cite any additional relevant literature that has been published since the original submission and mention any additional citations in your response. 

Before you revise your manuscript, please review the following PLOS policy and formatting requirements checklist PDF: http://journals.plos.org/plosbiology/s/file?id=9411/plos-biology-formatting-checklist.pdf. It is helpful if you format your revision according to our requirements - should your paper subsequently be accepted, this will save time at the acceptance stage.

Please note that as a condition of publication PLOS' data policy (http://journals.plos.org/plosbiology/s/data-availability) requires that you make available all data used to draw the conclusions arrived at in your manuscript. If you have not already done so, you must include any data used in your manuscript either in appropriate repositories, within the body of the manuscript, or as supporting information (N.B. this includes any numerical values that were used to generate graphs, histograms etc.). For an example see here: http://www.plosbiology.org/article/info%3Adoi%2F10.1371%2Fjournal.pbio.1001908#s5.

For manuscripts submitted on or after 1st July 2019, we require the original, uncropped and minimally adjusted images supporting all blot and gel results reported in an article's figures or Supporting Information files. We will require these files before a manuscript can be accepted so please prepare them now, if you have not already uploaded them. Please carefully read our guidelines for how to prepare and upload this data: https://journals.plos.org/plosbiology/s/figures#loc-blot-and-gel-reporting-requirements.

Upon resubmission, the editors will assess your revision and if the editors and Academic Editor feel that the revised manuscript remains appropriate for the journal, we will send the manuscript for re-review. We aim to consult the same Academic Editor and reviewers for revised manuscripts but may consult others if needed.

If you still intend to submit a revised version of your manuscript, please go to https://www.editorialmanager.com/pbiology/ and log in as an Author. Click the link labelled 'Submissions Needing Revision' where you will find your submission record. 

Sincerely,

Ines

--

Ines Alvarez-Garcia, PhD

Senior Editor

PLOS Biology

Carlyle House, Carlyle Road

Cambridge, CB4 3DN

+44 1223–442810

Reviewers' comments

Rev. 1:

This manuscript provides clear evidence that CaMKII in general and especially CaMKII beta is important for proper expression of cargo of dense core vesicles (DCV) including BDNF and chromogranins.

Experiments are well executed and conclusions justified (but limited).

Major Concerns:

1. My main concern is that the work is very focused and limited in scope. This manuscript would strongly profit if mechanistic insight into how CaMKII mediates its effect on cargo expression and secretory transport could be provided.

2. Fig. 2A: vesicular SEP-BDNF content was only quantified with NH4 after high frequency stimulation. That should be systematically done without such stimulation under basal conditions.

3. Fig. 2B: it seems the SEP-BDNF puncta density is comparable for lower densities but the higher densities seen in WT neurons are where DKO neurons fail to match-perhaps this distribution can be further analyzed and interpreted. Is there a limit/ceiling effect of maximal capacity for DCV density?

4. The authors state: “ the minor changes in Syb2, Munc18, and SNAP25 suggest that CaMKII has a minor role…” (top of p9). A 30% reduction in Munc18 and 23% reduction in SNAP25 is not minor and could perhaps should?) affect vesicle release.

Minor Concerns:

1. BDNF is sometimes misspelled in Figures (Fig. 4B, left; Fig. 5B bottom BNDF2).

Rev. 2: Kang Shen - please note that this reviewer has waived anonymity

Moro et al., studied the effect of CaMKIIs on dense core vesicle trafficking, release and neuropeptide expression in hippocampal neurons. They found that CaMKIIs increases the expression of multiple neuropeptide genes, most likely through regulation of transcription, an effect that is mostly carried out by the betaCaMKII. They also reported that the CaMKIIs increased the transport of DCVs into the axon but does not play a major effect on the release of DCVs. The strength of this manuscript lies in two folds. First they have taken the stringent genetic approach and examined the alpha and beta CaMKII double knockout cells. This clearly eliminated any specificity concerns from the pharmacology experiments and the rescue experiments demonstrated convincingly that the effects of these knockouts are due to CaMKIIs. Secondly, they have examined DCV release with single vesicle resolution which was not done in C. elegans. The weakness of the manuscript lies in the fact that the conceptual advance is rather limited. The regulation of CaMKII on BDNF transcription has been shown using drug experiments in mouse. The effect of CaMKII mutation in DCV abundance has been characterized in C. elegans. This manuscript confirmed these observations in mouse with a stringent experimental approach and showed that beta CaMKII is playing a more important role. However, the mechanisms of these effects are not explored. For example, examining the phosphorylation of CREB would be a potential way to know if the transcriptional effect is due to CREB. In the mammalian literature, CaMKIV has been more linked to CREB phosphorylation.

Below are my other technical comments.

In Fig. 1, is the fluorescence intensity before stimulation the same between wt and DKO?

In Fig. 4, only the frequency of DCV movements was quantified. The authors should also quantify the speed and pauses of these movements. These information will further provide some insight in what is causing the reduced DCV entry into axon.

Line 143 “our data demonstrate that CaMKII regulates neuromodulation by lowering the

number of available DCVs and that is not involved in the fusion of SVs or the mechanism of fusion of DCVs in hippocampal neurons.” Shouldn’t this be increasing the number of DCV?

Line 258 “Therefore, CaMKII does not regulate the trafficking of DCVs in the axon but not take part in the

259 stimulation dependent arrest of trafficking vesicles.” This sentence is problematic. Shouldn’t it be “CaMKII does regulate…”?

Rev. 3:

In this study, the authors investigated the necessity of alpha- and beta-CaMKII in the secretion of BDNF, a well-established neuromodulator in synapse development and plasticity using mouse alpha- and beta-CaMKII deficient, hippocampal cultured neurons. The findings were novel and interesting. However, the manuscript in the current form requires substantial data to solidify sufficiency or necessity of either alpha- or beta-CaMKII in general neuromodulator secretion, given that most of findings were derived from exogenous overexpression of BDNF pHluorin only. In addition, the authors may want to show significance of their findings in conjunction with synapse development and/or plasticity. Therefore, the authors should perform experiments to test whether the loss of alpha- and beta- CaMKII affects- 1) synaptic vesicle secretion using vGlut- or synaptobrevin-pHluorin (or electrophysiological recordings), 2) exocytosis of general cargos using transferrin-receptor-pHluorin from dendrites. These experiments will confirm the authors’ claim on the regulatory role of alpha- and beta- CaMKII on neuromodulators.

Minor points:

1. Is the number of BDNF puncta reduced in the distal dendrite of DKO neurons? Did the authors perform any detailed quantification of BDNF puncta in relation to their distances from the soma?

2. The authors may want to include recovery kinetics from FRAP experiments in Figure 4C and D.

3. BDNF was misspelled in multiple places (Figures 1G, 4B, and 5B).

4. Based on supplementary Fig 8, the overexpression of beta-CaMKII resulted in dramatic morphological changes. Why?

---

## [Decision Letter · Decision Letter 2]

23 Jun 2020

Dear Dr Verhage,

Thank you for submitting your revised Research Article entitled "CaMKII controls neuromodulation via neuropeptide gene expression and axonal targeting of neuropeptide vesicles" for publication in PLOS Biology. I have now obtained advice from the three original reviewers and have discussed their comments with the Academic Editor. 

We're delighted to let you know that we're now editorially satisfied with your manuscript. However before we can formally accept your paper and consider it "in press", we also need to ensure that your article conforms to our guidelines. A member of our team will be in touch shortly with a set of requests. As we can't proceed until these requirements are met, your swift response will help prevent delays to publication. Please also make sure to address the data and other policy-related requests noted at the end of this email.

*Copyediting*

*Published Peer Review History*

*Early Version*

*Submitting Your Revision*

Sincerely,

Ines

--

Ines Alvarez-Garcia, PhD

Senior Editor

PLOS Biology

Carlyle House, Carlyle Road

Cambridge, CB4 3DN

+44 1223–442810

DATA POLICY:

Thank you for sending us the file containing all the data underlying the graphs shown in the figures. I have checked the data and found a few mistakes and missing data that should be addressed: 

Fig. 3L – please correct label (at the moment it’s 3K)

Fig. 5B – please correct the label of BDNF4 data (at the moment it’s BDNF2)

Fig. S7 - please relabel (at the moment it’s S4) and add the data of Fig. S7C, C, G, H, D, E, J, K, M, N

Fig. S9A – please add the data

In addition, please indicate in each figure legend (including those from the Supplementary figures) WHERE THE UNDERLYING DATA CAN BE FOUND.

Reviewers’ comments

Rev. 1:

The Authors addressed the concerns well, including providing further mechanistic insight.

Rev. 2:

The new data added to this manuscript has significantly extended the scope for this paper. The authors have also addressed my questions. I have no more questions.

Rev. 3:

The authors addressed all of my concerns by performing additional experiments. I have no further comment.

---

## [Editor Report · Decision Letter 3]

17 Jul 2020

Dear Dr Verhage,

On behalf of my colleagues and the Academic Editor, Eunjoon Kim, I am pleased to inform you that we will be delighted to publish your Research Article in PLOS Biology. 

Early Version

PRESS 

Kind regards,

Vita Usova

Publication Assistant, 

PLOS Biology

on behalf of

Ines Alvarez-Garcia,

Senior Editor

PLOS Biology